# Antiviral function and viral antagonism of the rapidly evolving dynein activating adaptor NINL

**Donté Alexander Stevens[1†], Christopher Beierschmitt[2†], Swetha Mahesula[1,3], Miles R Corley[2], John Salogiannis[1‡], Brian V Tsu[2], Bryant Cao[2], Andrew P Ryan[2], Hiroyuki Hakozawki[4], Samara L Reck-Peterson[1,3,5]\*, Matthew D Daugherty[2]\***

[1]Department of Cellular and Molecular Medicine, University of California, San Diego, La Jolla, United States; [2]Department of Molecular Biology, University of California, San Diego, La Jolla, United States; [3]Howard Hughes Medical Institute, Chevy Chase, United States; [4]Nikon Imaging Center at UC San Diego, University of California, San Diego, San Diego, United States; [5]Department of Cell and Developmental Biology, University of California, San Diego, La Jolla, United States

**\*For correspondence:**
sreckpeterson@ucsd.edu (SLR-P);
mddaugherty@ucsd.edu (MDD)

[†]These authors contributed equally to this work

**Present address:** [‡]Department of Molecular Physiology and Biophysics, University of Vermont, Burlington, United States

**Abstract** Viruses interact with the intracellular transport machinery to promote viral replication. Such host–virus interactions can drive host gene adaptation, leaving signatures of pathogen-driven evolution in host genomes. Here, we leverage these genetic signatures to identify the dynein activating adaptor, ninein-like (NINL), as a critical component in the antiviral innate immune response and as a target of viral antagonism. Unique among genes encoding components of active dynein complexes, NINL has evolved under recurrent positive (diversifying) selection, particularly in its carboxy-terminal cargo-binding region. Consistent with a role for NINL in host immunity, we demonstrate that NINL knockout cells exhibit an impaired response to interferon, resulting in increased permissiveness to viral replication. Moreover, we show that proteases encoded by diverse picornaviruses and coronaviruses cleave and disrupt NINL function in a host- and virus-specific manner. Our work reveals the importance of NINL in the antiviral response and the utility of using signatures of host–virus genetic conflicts to uncover new components of antiviral immunity and targets of viral antagonism.

## Editor's evaluation

The findings of a new player in interferon signaling that is both rapidly evolving and antagonized by a viral protease are exciting.

## Introduction

Viruses interact directly with host proteins at nearly every step of their replication cycle. Such molecular interactions can either benefit the host (e.g., immune recognition) or the virus (e.g., viral co-option of host machinery or viral antagonism of host immunity) and are thus critical determinants of the outcome of a viral infection. Competing genetic innovations on both sides of these host–virus conflicts result in signatures of recurrent adaptation that have been described as molecular 'arms races' (*Daugherty and Malik, 2012*; *Duggal and Emerman, 2012*; *Tenthorey et al., 2022*). Indeed, many host antiviral factors that directly interact with viral proteins display signatures of recurrent positive (diversifying) selection over evolutionary time, and genetic variation in these host–virus interactions shapes species-specific susceptibility to circulating and emerging pathogens (*Daugherty and*

**eLife digest** Humans and viruses are locked in an evolutionary arms race. Viruses hijack cells, using their resources and proteins to build more viral particles; the cells fight back, calling in the immune system to fend off the attack. Both actors must constantly and quickly evolve to keep up with each other. This genetic conflict has been happening for millions of years, and the indelible marks it has left on genes can serve to uncover exactly how viruses interact with the organisms they invade.

One hotspot in this host-virus conflict is the complex network of molecules that help to move cargo inside a cell. This system transports elements of the immune system, but viruses can also harness it to make more of themselves. Scientists still know very little about how viruses and the intracellular transport machinery interact, and how this impacts viral replication and the immune response.

Stevens et al. therefore set out to identify new interactions between viruses and the transport system by using clues left in host genomes by evolution. They focused on dynein, a core component of this machinery which helps to haul molecular actors across a cell. To do so, dynein relies on adaptor molecules such as 'Ninein-like', or NINL for short.

Closely examining the gene sequence for NINL across primates highlighted an evolutionary signature characteristic of host-virus genetic conflicts; this suggests that the protein may be used by viruses to reproduce, or by cells to fend off infection.

And indeed, human cells lacking the NINL gene were less able to defend themselves, allowing viruses to grow much faster than normal. Further work showed that NINL was important for a major type of antiviral immune response. As a potential means to sabotage this defence mechanism, some viruses cleave NINL at specific sites and disrupt its role in intracellular transport.

Better antiviral treatments are needed to help humanity resist old foes and new threats alike. The work by Stevens et al. demonstrates how the information contained in host genomes can be leveraged to understand what drives susceptibility to an infection, and to pinpoint molecular actors which could become therapeutic targets.

---

*Malik, 2012*; *Duggal and Emerman, 2012*; *Tenthorey et al., 2022*; *Meyerson and Sawyer, 2011*; *Rothenburg and Brennan, 2020*). These data suggest that there is great potential to use evolutionary signatures of rapid evolution not only as an approach to more deeply understand known host–virus conflicts but also as a means to discover new proteins engaged in viral interactions (*Daugherty and Malik, 2012*). Compellingly, it is estimated that around 30% of all adaptive amino acid changes in humans result from viral selective pressure (*Enard et al., 2016*; *Enard and Petrov, 2018*), suggesting that many host–virus conflicts remain undescribed.

One potential source of host–virus conflicts is over the active transport of macromolecules within the cell. The relatively large size of eukaryotic cells, coupled with the density of macromolecules in the cytoplasm, limits the effectiveness of diffusion to localize and transport large intracellular components, such as organelles, membrane vesicles, RNAs, and protein complexes (*Luby-Phelps, 2000*; *Seksek et al., 1997*). Eukaryotic cells overcome this problem by actively transporting large intracellular cargos using dynein and kinesin motors, which move on microtubules in opposite directions. Aspects of viral infection, viral replication, and the host immune response all require microtubule-based transport. For example, viruses co-opt the microtubule cytoskeleton for cell entry, transport of viral components to sites of replication, remodeling of cellular compartments, and viral egress (*Brandenburg and Zhuang, 2007*; *Dodding and Way, 2011*; *Döhner et al., 2005*; *Radtke et al., 2006*). Similarly, in response to infection, the host adaptive and innate immune response require movement of signaling components, transport of endocytic and exocytic vesicles, organelle recycling, and cellular remodeling, all of which require the microtubule-based trafficking machinery (*Ilan-Ber and Ilan, 2019*; *Kast and Dominguez, 2017*; *Man and Kanneganti, 2016*; *Mostowy and Shenoy, 2015*). Despite the clear role of microtubule-based transport in both aiding and inhibiting viral replication, the degree to which host–virus genetic conflicts shape the basic biology of this machinery is poorly understood.

To uncover the role of this machinery in viral replication and the immune response, we set out to determine whether there were undescribed genetic conflicts between viruses and the intracellular transport machinery. Specifically, we focused on the dynein transport machinery, which traffics dozens of cellular cargos towards microtubule minus-ends (generally anchored to centrosomes near the

nucleus). In human cells, only one dynein motor-containing gene, cytoplasmic dynein-1 (*DYNC1H1*), is responsible for long-distance transport in the cytoplasm. However, the active cytoplasmic dynein-1 complex (hereafter dynein) is composed of multiple dynein subunits, the multisubunit dynactin complex, and one of a growing list of activating adaptors (*McKenney et al., 2014*; *Schlager et al., 2014*). Interestingly, it is the activating adaptors that provide cargo specificity for dynein in addition to their essential role in activating robust processive motility (*Olenick and Holzbaur, 2019*; *Reck-Peterson et al., 2018*). However, the specific biological functions of most activating adaptors remain unknown.

We now show that one activating adaptor, ninein-like protein (NINL, also known as NLP), has evolved under recurrent positive selection in primates, making it unique among all analyzed dynein, dynactin, and activating adaptor genes. Based on the hypothesis that such an evolutionary signature in NINL could be the result of a previously undescribed host–virus interaction, we explored the function of NINL in the antiviral immune response. Using NINL knockout (KO) cells, we find that NINL is important for limiting viral infection, especially in the presence of the antiviral signaling cytokine, type I interferon (IFN). We further demonstrate that this attenuation of the antiviral efficacy of IFN in NINL KO cells is due to a dramatic reduction in interferon-stimulated gene (ISG) production, likely as a consequence of decreased nuclear localization of phosphorylated STAT1 following IFN treatment. Finally, we show that diverse proteases from picornaviruses and coronaviruses cleave NINL at several host-specific sites, resulting in a disruption of NINL's ability to traffic cargo. Together, our results reveal a novel immune function for NINL as well as a means by which viruses may antagonize NINL function in a virus- and host-specific manner. More broadly, our work implicates a component of the dynein transport machinery as a rapidly evolving barrier to viral replication and highlights the utility of an evolution-guided approach for discovery of new host–virus interactions and the genetic conflicts that may arise from them.

## Results
### The dynein activating adaptor, NINL, has evolved under positive selection

Active dynein complexes in human cells are large, multisubunit machines. The dynein/dynactin complex is composed of two copies of the ATPase-containing heavy chain, two copies of five additional dynein chains, the 23-subunit dynactin complex, and an activating adaptor (*Olenick and Holzbaur, 2019*; *Reck-Peterson et al., 2018*; *Figure 1A*). To generate hypotheses about potential genetic conflicts between the dynein machinery and pathogens, we searched for signatures of positive selection during primate evolution in genes for all dynein/dynactin subunits and the 13 activating adaptors known at the time of this analysis. Each human dynein gene was compared to orthologs in 13–20 diverse simian primates, and a gene-wide dN/dS (also known as omega) value was calculated, which compares the gene-wide rate of nonsynonymous changes (i.e., amino acid altering) to the rate of synonymous (i.e., silent) changes. Consistent with the critical role of dynein-mediated intracellular transport, most genes we analyzed were extremely well conserved with dN/dS values of <0.1, while one dynein activating adaptor, NINL, showed an elevated rate compared to the rest (*Figure 1B* and *Supplementary file 1*). To determine whether any genes had individual codons that have been subject to recurrent positive selection, we performed codon-based analyses of positive selection. Consistent with their low dN/dS values, we observed that most dynein, dynactin, and activating adaptor genes showed no evidence for positive selection (p-value>0.05). In contrast, NINL showed strong evidence for recurrent positive selection in primates, consistent with previous data (*Gordon et al., 2020*; *Figure 1B* and *Supplementary file 1*), establishing the possibility that NINL could be at the interface of a host–pathogen interaction.

In order to attribute the signatures of positive selection in NINL to known functional domains within NINL, we performed additional analyses to identify specific codons that have evolved under positive selection using three independent methods: PAML, FEL, and MEME (*Kosakovsky Pond and Frost, 2005*; *Murrell et al., 2012*; *Yang, 2007*). We identified 30 codons that show signatures of positive selection based on one or more of these methods (*Figure 1C* and *Supplementary file 2*). Most (24 of 30) of these codons are excluded from the known dynein/dynactin-binding region of NINL (residues 1–702) (*Redwine et al., 2017*) and instead are located in the carboxy-terminal region of the protein

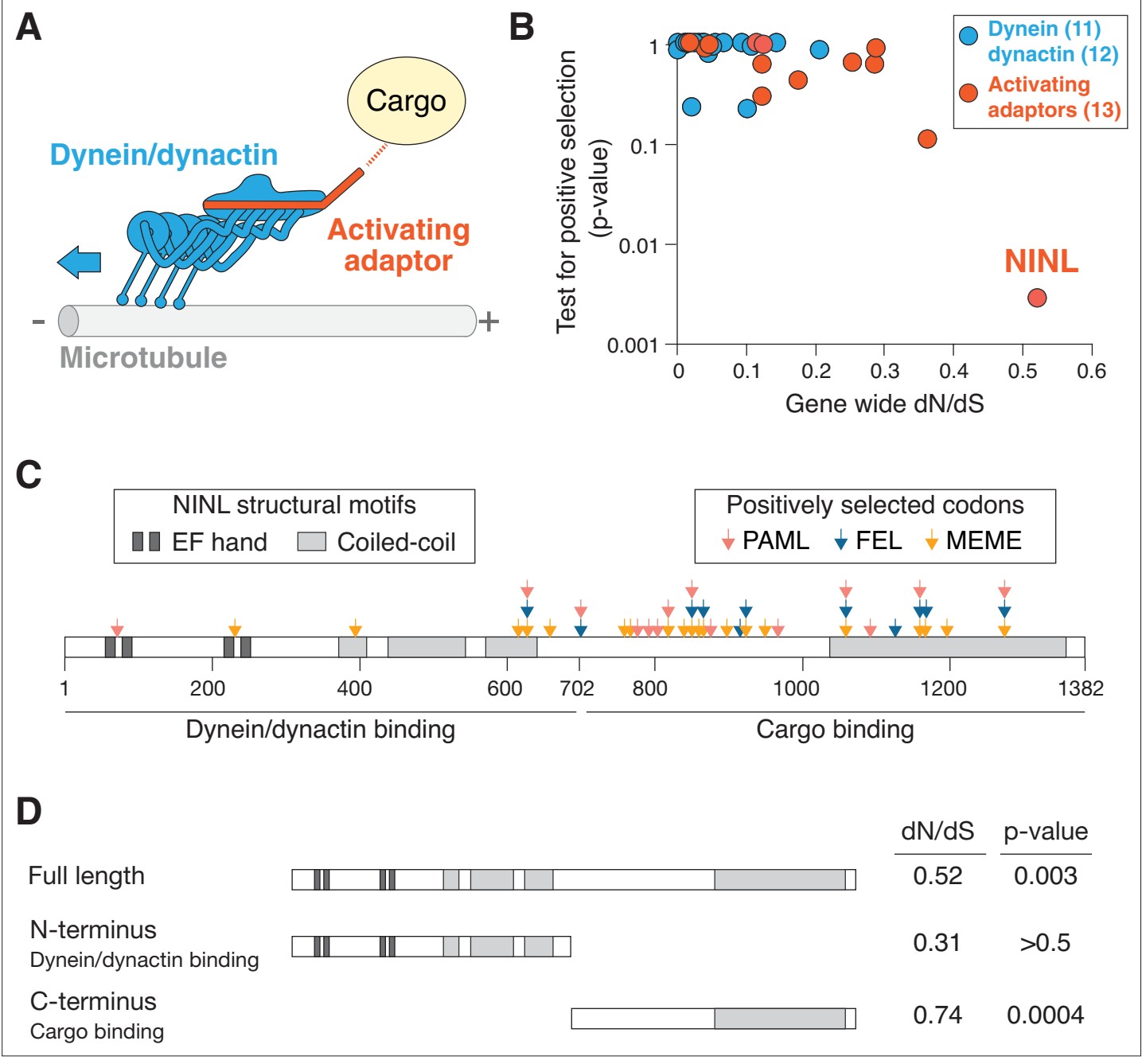

**Figure 1.** The dynein activating adaptor, ninein-like (NINL), has evolved under positive selection in primates. (**A**) A schematic of the cytoplasmic dynein-1 transport machinery, which includes dynein and dynactin subunits (blue) and an activating adaptor (orange). Dynein moves toward the minus-end of microtubules (blue arrow). (**B**) A scatterplot displaying evolutionary signatures of selection for 23 dynein and dynactin genes (blue) and 13 dynein activating adaptor genes (orange). The x-axis displays the rate of nonsynonymous changes (dN) divided by the rate of synonymous changes (dS) in the coding sequence across primate evolution. The y-axis displays the calculated probability of the gene having evolved under positive selection using PAML. Complete data are found in *Supplementary file 1* . (**C**) A schematic of human NINL isoform 1 with EF hand (dark gray) and coiled-coil (light gray) domains shown. The amino-terminal dynein/dynactin-binding region and the carboxy-terminal candidate cargo-binding domains are indicated. Sites of positive selection predicted by three evolutionary models are shown as colored arrows: PAML (light red), FEL (blue), and MEME (orange). A full list of sites and their calculated probabilities is shown in *Supplementary file 2* . (**D**) Full-length NINL, the dynein/dynactin-binding amino-terminus of NINL and the candidate cargo-binding carboxy-terminus of NINL were analyzed for signatures of positive selection. Select dN/dS and p-values are shown, with additional evolutionary data in *Supplementary file 3*.

that is expected to interact with cargo (residues 703–1382). When we analyzed individual domains on their own, we found no evidence for positive selection in the amino-terminus alone, while the carboxy-terminus retained a significant signature of positive selection (*Figure 1D* and *Supplementary file 3*). Taken together, our evolutionary analyses indicate that NINL stands out among components of the active dynein complex by having evolved under recurrent positive selection in primates.

## Viral replication is increased in cells lacking NINL

Our observation that NINL displays a signature of positive selection that is unique among dynein components led us to hypothesize that NINL may be co-opted by viruses for viral replication or may play a role in the immune response to viruses. To evaluate this hypothesis, we generated a human A549 cell line that lacked NINL (NINL KO) (*Figure 2A*, *Figure 2—figure supplement 1A and B*) as A549 cells are susceptible to infection with many viruses and mount an effective antiviral response after treatment with IFN. At a qualitative level, these cells appeared to have a normal microtubule architecture and centrosomes (*Figure 2—figure supplement 1A*). In parallel, we generated cells that lacked ninein (NIN KO) (*Figure 2A*, *Figure 2—figure supplement 1C*), the closest human paralog to NINL, which shares a similar domain architecture with NINL and is also a dynein activator adaptor (*Redwine et al., 2017*), but shows no evidence for positive selection (*Figure 1B* and *Supplementary file 1*). To evaluate the effect that NINL or NIN have on viral replication or the innate immune response to viral infection, we infected WT, NINL KO, or NIN KO A549 cells with a model enveloped negative-sense single-stranded RNA (-ssRNA) virus, vesicular stomatitis virus (VSV), with and without pretreatment with the antiviral signaling cytokine interferon alpha (IFNα). Consistent with the strong antiviral effect of IFNα, we observed a >100-fold decrease in viral replication in WT and NIN cells that had been pretreated with IFNα (*Figure 2B*). In contrast, we observed that the effect of IFNα was significantly attenuated in NINL KO cells, where we found that IFNα pretreatment reduced VSV replication <10-fold (*Figure 2B*). To attribute the changes in viral replication to the absence of NINL rather than off-target perturbations or cell-line specific effects, we generated additional NINL KO cell lines in human U-2 OS cells (*Figure 2—figure supplement 1D and E*). We again observed that NINL KO cells had a significant reduction in the antiviral effects of IFNα pretreatment (*Figure 2C*), although the magnitude of the IFN effect varied between cell types likely due to cell type-specific differences in basal and induced ISG expression (*Rusinova et al., 2013*). We also noted that VSV replication was higher in NINL KO cell lines compared to WT even in the absence of IFN, which may indicate either a basal defect in the antiviral response in NINL KO cells or a second function of NINL that is IFN independent. To test whether the NINL-dependent effect on IFN antiviral potency was specific to VSV replication, we tested two positive-sense single-stranded RNA (+ssRNA) viruses: Sindbis virus (SinV) – an enveloped virus – and coxsackievirus B3 (CVB3) – a non-enveloped virus in both A549 and U-2 OS cells. Each virus was sensitive to the antiviral effect of IFN in WT and NIN KO cells, although the degree of sensitivity was different for each virus as would be expected due to known differences in viral sensitivity to different ISGs (*Schoggins et al., 2014*; *Figure 2D*, *Figure 2—figure supplement 2A–D*). However, with both SinV and CVB3, as we observed with VSV, the antiviral effect of IFN was reduced in NINL KO cells compared to WT or NIN KO cells (*Figure 2D*, *Figure 2—figure supplement 2A–D*). The attenuation of the IFN-induced antiviral effect against viruses from three distinct families suggests that NINL may broadly play a role in the IFN-mediated innate immune response to viruses.

## Loss of NINL results in an attenuated antiviral immune response

Based on the reduced antiviral potency of IFNα in cells lacking NINL, we next investigated whether there was an attenuation of IFN-mediated signaling in NINL KO cells. Type I IFNs, such as IFNα, activate the Janus kinase/signal transducer and activator of transcription (JAK/STAT) pathway to trigger the expression of ISGs, which include potent antiviral effectors (*Schoggins, 2019*). Therefore, we asked whether there was a defect in the JAK/STAT signaling cascade by western blot analysis of the phosphorylation of the transcription factors STAT1 (pSTAT1(Y701)) and STAT2 (pSTAT2(Y690)) as well as the induction of ISG expression following IFNα pretreatment. Despite robust phosphorylation of STAT1 and STAT2 in response to IFNα pretreatment in WT, NINL KO, and NIN KO cells, protein expression of the canonical ISGs–MX1, IFIT3, OAS1, and ISG15 was greatly reduced in NINL KO cells relative to WT or NIN KO cells (*Figure 3A*). To again confirm that this was not specific to cell type, we showed that this lack of ISG protein expression was independent of cell background or the choice

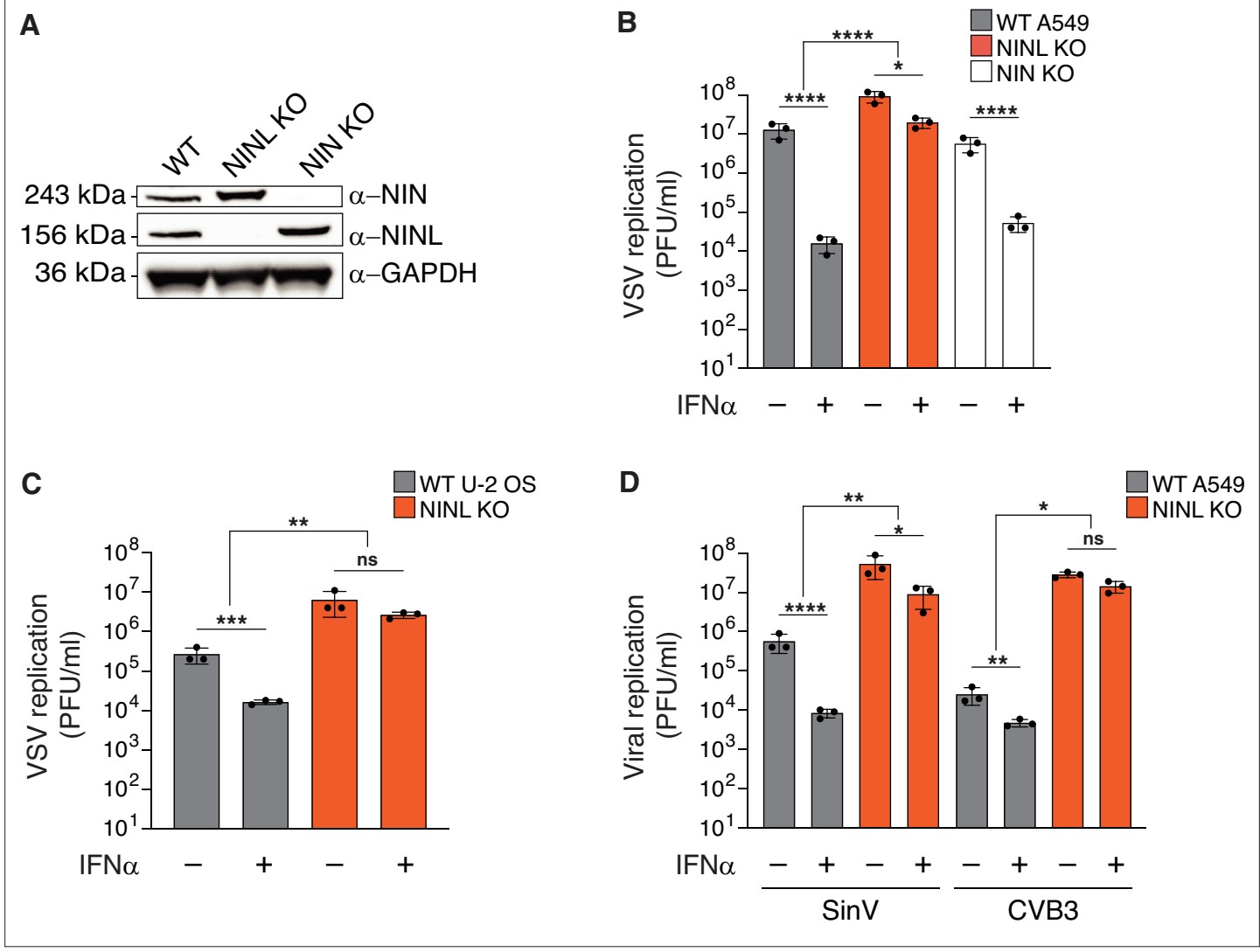

**Figure 2.** The antiviral potency of interferon alpha (IFNα) is reduced in ninein-like (NINL) knockout (KO) cells. (**A**) Immunoblots of wild-type (WT) A549 cells, and CRISPR/Cas9-generated NINL and NIN KO A549 cells probed with the indicated antibodies. GAPDH served as a loading control. Protein molecular weight markers are shown in kilodaltons (kDa) to the left of each immunoblot. Representative images from three biological replicates are shown. (**B**) WT, NINL KO, and NIN KO A549 cells were treated with 100U IFNα for 24 hr and then infected with vesicular stomatitis virus (VSV) (5000 PFU/mL, MOI ≈ 0.01). Virus-containing supernatants were collected 9 hr post-infection and viral titers (y-axis, PFU/mL) were determined by plaque assay. (**C**) WT or NINL KO U-2 OS cells were treated with 100U IFNα for 24 hr and then infected with VSV (5000 PFU/mL, MOI ≈ 0.01). Virus-containing supernatant was collected 9 hr post-infection and viral titers (y-axis, PFU/mL) were determined by plaque assay. (**D**) WT or NINL KO A549 cells were treated with 100U IFNα for 24 hr and then infected with Sindbis virus (500,000 PFU/mL, MOI ≈ 1.0) (left) or treated with 1000U IFNα for 24 hr and then infected with coxsackievirus B3 (5000 PFU/mL, MOI ≈ 0.01) (right). Virus-containing supernatants were collected 24 hr post-infection and viral titers (y-axis, PFU/mL) were determined by plaque assay. (**B–D**) Data are presented as mean ± standard deviation of three experiments, with individual points shown. Data were analyzed by two-way ANOVA with Tukey's method adjustment for multiple comparisons for IFNα treatment within each cell line, two-way ANOVA interaction comparison for IFNα interaction between cell lines. *$p < 0.05$, **$p < 0.01$, ***$p < 0.001$, ****$p < 0.0001$, ns, not significant.

The online version of this article includes the following source data and figure supplement(s) for figure 2:

**Source data 1.** Full raw unedited images for *Figure 2A*.

**Source data 2.** Individual data values for *Figure 2B–D*.

**Figure supplement 1.** Validation of CRISPR/Cas9-editing to generate ninein-like (NINL) and NIN knockout (KO) cells.

**Figure supplement 1—source data 1.** Full raw unedited images for *Figure 2—figure supplement 1D*.

**Figure supplement 2.** Reduction of interferon alpha (IFNα)-mediated antiviral response is observed across multiple cell lines.

**Figure supplement 2—source data 1.** Individual data values for *Figure 2—figure supplement 2A–D*.

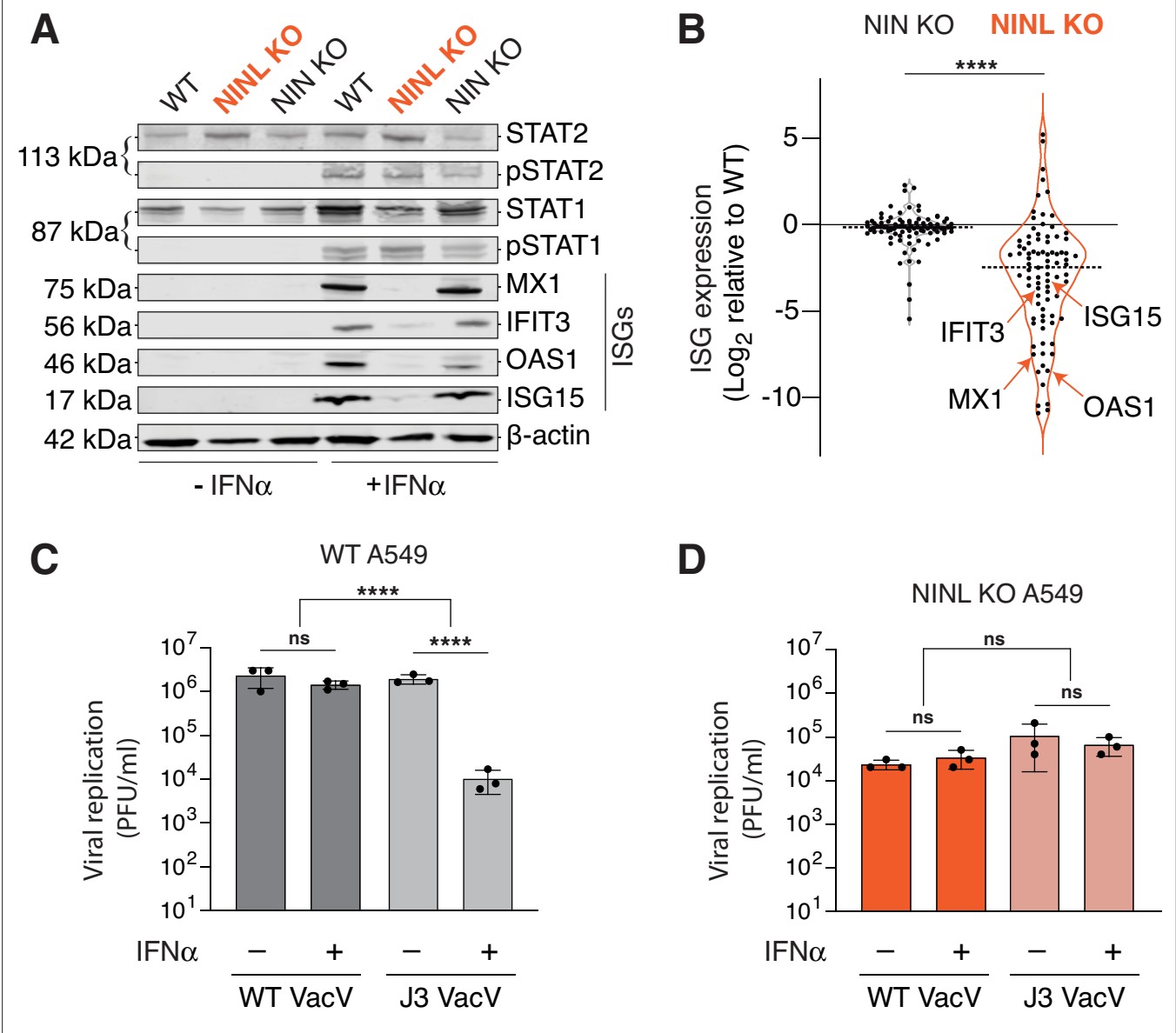

**Figure 3.** Ninein-like (NINL) knockout (KO) cells fail to mount an effective interferon (IFN) response. (**A**) Immunoblot of extracts from wild-type (WT), NINL KO, and NIN KO A549 cells untreated (-) or treated (+) with IFNα. Immunoblots were probed with anti-STAT2, anti-phospho-STAT2 (Tyr690), anti-STAT1, anti-phospho-STAT1 (Tyr701), anti-Mx1, anti-IFIT3, anti-OAS1, anti-ISG15, and anti-β-actin antibodies. Predicted protein molecular weights are shown in kilodaltons (kDa) to the left of each immunoblot. Representative images from three biological replicates are shown. (**B**) Differential interferon-stimulated gene (ISG) expression in WT, NINL KO, and NIN KO cells induced with IFNα. ISGs were identified as the 88 genes whose expression was upregulated in WT cells after IFNα pretreatment (*Figure 3—figure supplement 2*). Data are displayed as a violin plot of ISG expression in NIN KO or NINL KO cells relative to WT cells. ****p<0.0001 based on paired *t*-test. Dotted line indicates mean. Individual datapoints for ISGs shown in panel (**A**) are indicated. (**C**) A549 WT cells were treated with 1000U IFNα for 24 hr, then infected with wild-type vaccinia virus (WT VacV) or J3 mutant vaccinia virus (J3 VacV) (50,000 PFU/mL, MOI ≈ 0.1). Cell-associated virus was collected 24 hr post-infection and viral titers (y-axis, PFU/mL) were determined by plaque assay. (**D**) A549 NINL KO cells were treated, infected, harvested, and quantified as described in (**C**). (**C, D**) Data are presented as mean ± standard deviation of three experiments, with individual points shown. Data were analyzed by two-way ANOVA with Tukey's method adjustment for multiple comparisons for IFNα treatment within each cell line, two-way ANOVA interaction comparison for IFNα interaction between cell lines. ****p<0.0001, ns, not significant.

The online version of this article includes the following source data and figure supplement(s) for figure 3:

**Source data 1.** Full raw unedited images for *Figure 3A*.

**Source data 2.** Individual data values for *Figure 3C*.

*Figure 3 continued on next page*

*Figure 3 continued*

**Figure supplement 1.** Reduced interferon-stimulated gene (ISG) production occurs following ninein-like (NINL) knockout (KO) in multiple cell lines generated using different CRISPR gRNAs.

**Figure supplement 1—source data 1.** Full raw unedited images for *Figure 3—figure supplement 1*.

**Figure supplement 2.** Identification of 88 interferon-stimulated genes (ISGs) in wild-type (WT) A549 cells.

**Figure supplement 3.** Interferon induction has a reduced effect on interferon-stimulated gene (ISG) expression in ninein-like (NINL) knockout (KO) cells.

**Figure supplement 4.** Differential gene expression in ninein-like (NINL) knockout (KO) and NIN KO cells compared to wild-type (WT) cells.

**Figure supplement 5.** Ninein-like (NINL) knockout (KO) results in loss of interferon sensitivity of the VacV J3 mutant.

**Figure supplement 5—source data 1.** Individual data values for *Figure 3—figure supplement 5A–C*.

of CRISPR guide (*Figure 2—figure supplement 1A–G*, *Figure 3—figure supplement 1*). Next, we performed RNAseq analyses on WT, NINL KO, and NIN KO A549 cells in the presence or absence of IFNα pretreatment (*Supplementary file 4*). In WT cells, we identified 88 ISGs that were significantly (adjusted p-value≤0.05, $\log_2$-fold change ≥ 1) upregulated in response to IFN treatment (*Figure 3—figure supplement 2*). We then compared the transcriptional profiles of these ISGs between IFNα-treated WT, NINL KO, and NIN KO cells. Consistent with our western blot analysis, the induction pattern of ISG transcripts in WT and NIN KO cells was similar, whereas many ISG transcripts from IFNα-treated NINL KO cells were downregulated compared to IFNα-treated WT cells (*Figure 3B*, *Figure 3—figure supplement 3*, and *Figure 3—figure supplement 4*). Other transcripts unrelated to the IFN response also showed altered expression in NINL KO cells relative to WT cells (*Figure 3—figure supplement 4* and *Supplementary file 4*). However, the overall lower expression of ISGs in NINL KO relative to WT cells (*Figure 3—figure supplement 4*) indicates that cells lacking NINL have a distinct defect in their ability to mount an effective antiviral immune response.

To further demonstrate that the lack of ISG expression in cells lacking NINL has a profound effect on the interferon-mediated antiviral response, we took advantage of a virus in which interferon sensitivity can be modulated genetically. Vaccinia virus (VacV) is a large double-stranded DNA virus that is relatively insensitive to the effects of IFNα due to the large number of proteins the virus encodes that antagonize the immune response (*Yu et al., 2021*). However, a point mutation in the J3 methyltransferase protein (VacV J3) confers interferon sensitivity by sensitizing the virus to the antiviral effects of the IFIT family of ISGs (*Daffis et al., 2010*; *Daugherty et al., 2016*; *Johnson et al., 2018*; *Latner et al., 2002*). As IFIT1, IFIT2, and IFIT3 were among the ISGs we saw decreased in NINL KO cells relative to WT and NIN KO cells (*Supplementary file 4*), we hypothesized that NINL KO cells may lack the ability to inhibit the J3 mutant VacV after IFNα pretreatment. As expected, in WT and NIN KO A549 cells, wild-type VacV (VacV WT) was insensitive to IFNα, whereas VacV J3 replication was significantly reduced upon IFNα pretreatment (*Figure 3C*, *Figure 3—figure supplement 5A*). In contrast, mutant and wild-type viruses replicated equivalently in the NINL KO cells regardless of IFNα pretreatment (*Figure 3D*). We found the same differential response to the J3 VacV mutant between WT and NINL KO cells in U-2 OS cells, showing that this phenotype is not cell-type specific (*Figure 3—figure supplement 5*). Altogether, these data suggest that NINL plays a critical role during the IFN-mediated antiviral immune response, further substantiating our hypothesis that NINL is at the center of an antagonistic host–pathogen interaction.

## NINL KO cells have reduced pSTAT1 nuclear localization

Based on our observations that NINL KO cells have an attenuated IFN-mediated antiviral immune response, we next sought to determine where in the IFN signaling cascade NINL acts. Upon initiation of JAK/STAT signaling, STAT1 and STAT2 are phosphorylated, heterodimerize, and bind to IFN regulatory factor 9 (IRF9) to form the heterotrimeric interferon-stimulated gene factor 3 (ISGF3) (*Schneider et al., 2014*). ISGF3 then translocates to the nucleus and binds to the IFN-stimulated response elements (ISREs) found in the promoters of ISGs (*Schneider et al., 2014*). Thus far, our data suggest that NINL acts after the phosphorylation of STAT1 and STAT2 (*Figure 3A*), but before ISG transcription (*Figure 3B*). To determine whether pSTAT1 translocates to the nucleus normally in NINL KO cells, we next examined the localization of pSTAT1 in WT, NIN, and NINL KO A549 cells. In both WT and NIN KO cells, pretreatment with IFNα led to the translocation of pSTAT1 to the nucleus as expected (*Figure 4A*). In contrast, in NINL KO cells we observed diffuse cytoplasmic staining of

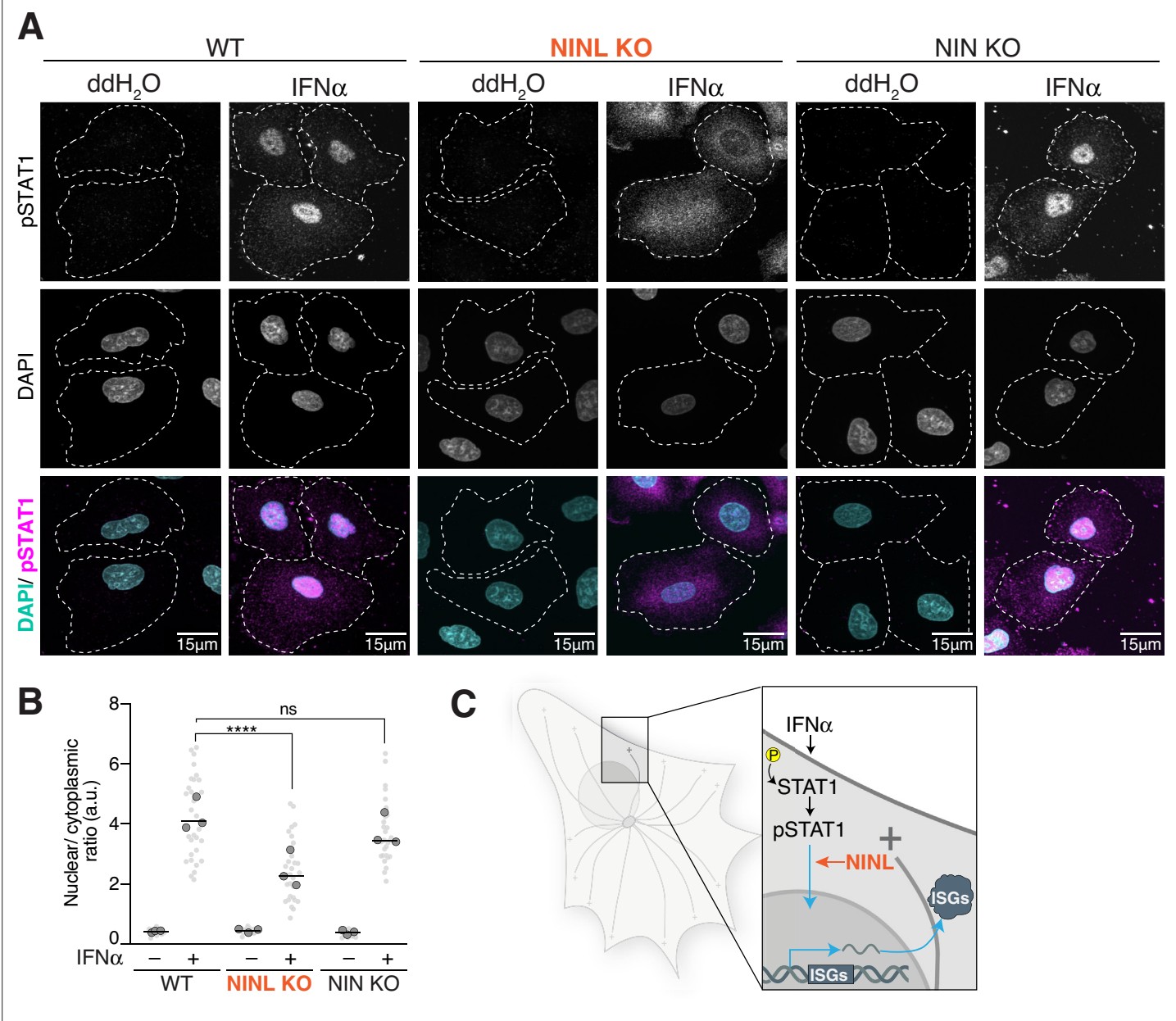

**Figure 4.** Ninein-like (NINL) knockout (KO) cells have reduced pSTAT1 nuclear localization. (**A**) Confocal micrographs displayed as maximum intensity projections of wild-type (WT), NINL KO, and NIN KO A549 cells. Where indicated, cells were treated for 1 hr with 1000U interferon alpha (IFNα). pSTAT1 was immunostained with anti-phosphorylated STAT1 (Y701) and nuclei were visualized with DAPI. Dashed white lines denote cell boundaries. Representative micrographs from three biological replicates are shown. 15 µm scale bars are also shown. (**B**) Quantification of the mean nuclear to mean cytoplasmic fluorescence intensity of pSTAT1 from three biological replicates. Light gray datapoints correspond to the mean ratio of all cells in a field of view. Dark gray, large, outlined circles correspond to the mean for each of the three biological replicates. For each condition, n = 30. The mean across all replicates is denoted by the bold line. Data were analyzed using Kruskal–Wallis with Dunn's post-hoc test for multiple comparisons. ****p<0.0001, ns, not significant. (**C**) Schematic depicting where in the IFN signaling pathway NINL acts.

The online version of this article includes the following source data and figure supplement(s) for figure 4:

**Source data 1.** Individual data values for *Figure 4B*.

**Figure supplement 1.** Ninein-like (NINL) knockout (KO) cells show reduced nuclear, but not cytoplasmic, intensity of pSTAT1.

**Figure supplement 1—source data 1.** Individual data values for *Figure 4—figure supplement 1*.

pSTAT1 after treatment with IFNα (*Figure 4A*). To quantify this observation, we measured the mean nuclear and cytoplasmic fluorescence intensity of pSTAT1 (*Figure 4—figure supplement 1A and B*). We then plotted the ratio of nuclear to cytoplasmic mean fluorescence intensity (*Figure 4B*). This analysis revealed that pSTAT1 shows significantly reduced nuclear localization in NINL KO cells (*Figure 4B*, *Figure 4—figure supplement 1B*). These data implicate NINL in the trafficking of the pSTAT1 transcription factor to the nucleus (*Figure 4C*).

## Viral proteases cleave NINL in a host-specific manner

The IFN response is the first line of host antiviral defense during viral infection. Thus, viruses have developed many strategies to evade or subvert the host IFN response (*Beachboard and Horner, 2016*; *Hoffmann et al., 2015*). Our data indicating that NINL is important for the IFN response, combined with the observation that NINL is evolving under positive selection, led us to hypothesize that viruses may antagonize NINL function. One such viral antagonism strategy is to deploy virus-encoded proteases to cleave components of the host antiviral defense system (*Lei and Hilgenfeld, 2017*; *Tsu et al., 2021b*). Thus, we next investigated whether viral proteases cleave NINL. Using a predictive model of enteroviral 3C protease (3C^pro) target specificity (*Tsu et al., 2021a*), we identified three high-confidence sites of potential cleavage within NINL at residues 231, 827, and 1032 (*Figure 5A*) in which amino acid diversity within primates is expected to alter 3C^pro cleavage susceptibility (*Figure 5B*). Indeed, upon transfection of NINL and CVB3 3C^pro into HEK293T cells, which were used for their ease of transfection and protein expression, we observed an overall reduction of full-length NINL and the appearance of two cleavage products at sizes that correspond to predicted cleavage at sites 827 and 1032 (*Figure 5C and D*). We also observed a weaker product at a size that corresponds to the predicted size of NINL after cleavage at site 231. To confirm cleavage site specificity, we generated NINL point mutants that take advantage of the diversity of these sites found in primates (*Figure 5B*). Specifically, we replaced the glutamine immediately preceding the site of cleavage (the P1 position) with an arginine found in non-human primates that we predicted would prevent cleavage by 3C^pros (*Tsu et al., 2021a*) for each of the predicted sites. Co-transfection of CVB3 3C^pro with NINL containing these mutations individually (Q1032R) or in combination (double mutant Q827R/Q1032R [Double] and triple mutant Q231R/Q827R/Q1032R [Triple]) confirmed the sites of cleavage, with the NINL triple mutant eliminating all cleavage products by CVB3 3C^pro (*Figure 5D*). We also noted that two of these sites (Q827 and Q1032), along with many of the codons predicted to be evolving under positive selection (*Figure 1C*), reside in a single exon (exon 17) within the carboxy-terminal region of NINL (*Figure 5A*). Intriguingly, this exon is lacking in an alternatively spliced isoform of NINL (isoform 2) (*Dona et al., 2015*; *Kersten et al., 2012*; *van Wijk et al., 2009*). We, therefore, tested whether isoform 2 is cleaved by CVB3 3C^pro. Consistent with the loss of two primary sites of cleavage, we observed minimal decrease in the full-length product when isoform 2 was co-transfected with CVB3 3C^pro, although we did observe weak protease-mediated cleavage at site 231 in isoform 2 (*Figure 5D*).

We next sought to understand the degree to which cleavage of NINL is conserved across viral proteases. We, therefore, tested a panel of 3C^pros from diverse viruses in the Picornaviridae family (*Tsu et al., 2021a*). Interestingly, while we found that all proteases tested were able to cleave NINL to some degree, the strength and position of cleavage was variable, even among proteases from closely related viruses such CVB3, enterovirus 71 (EV71), poliovirus (PV1), enterovirus D68 (EV68), and human rhinovirus A (HRVA), all of which are members of the enterovirus genus (*Figure 5—figure supplement 1A*). We also tested a panel of 3C-like proteases (3CL^pros) from members of the Coronaviridae family, including proteases from the betacoronaviruses, SARS-CoV-2 and SARS-CoV, and an alphacoronavirus, NL63-CoV. We again observed numerous cleavage products, some of which map to residues 827 and 1032 (*Figure 5—figure supplement 1B*), consistent with 3C^pros and 3CL^pros having similar active sites and cleavage preferences (*Ng et al., 2021*). Together, these data indicate that NINL is cleaved at species-specific sites by various proteases from human viruses. Such host- and virus-specificity of cleavage is a hallmark of host–virus arms races (*Tsu et al., 2021b*), further supporting the model that NINL's role in the interferon response positions it in evolutionary conflict with viruses.

We next aimed to confirm that infection-mediated cleavage efficiency and specificity recapitulated results we observed from transiently transfected viral proteases. We therefore infected cells expressing WT NINL and the NINL triple mutant with CVB3, a virus that encodes a 3C^pro that strongly

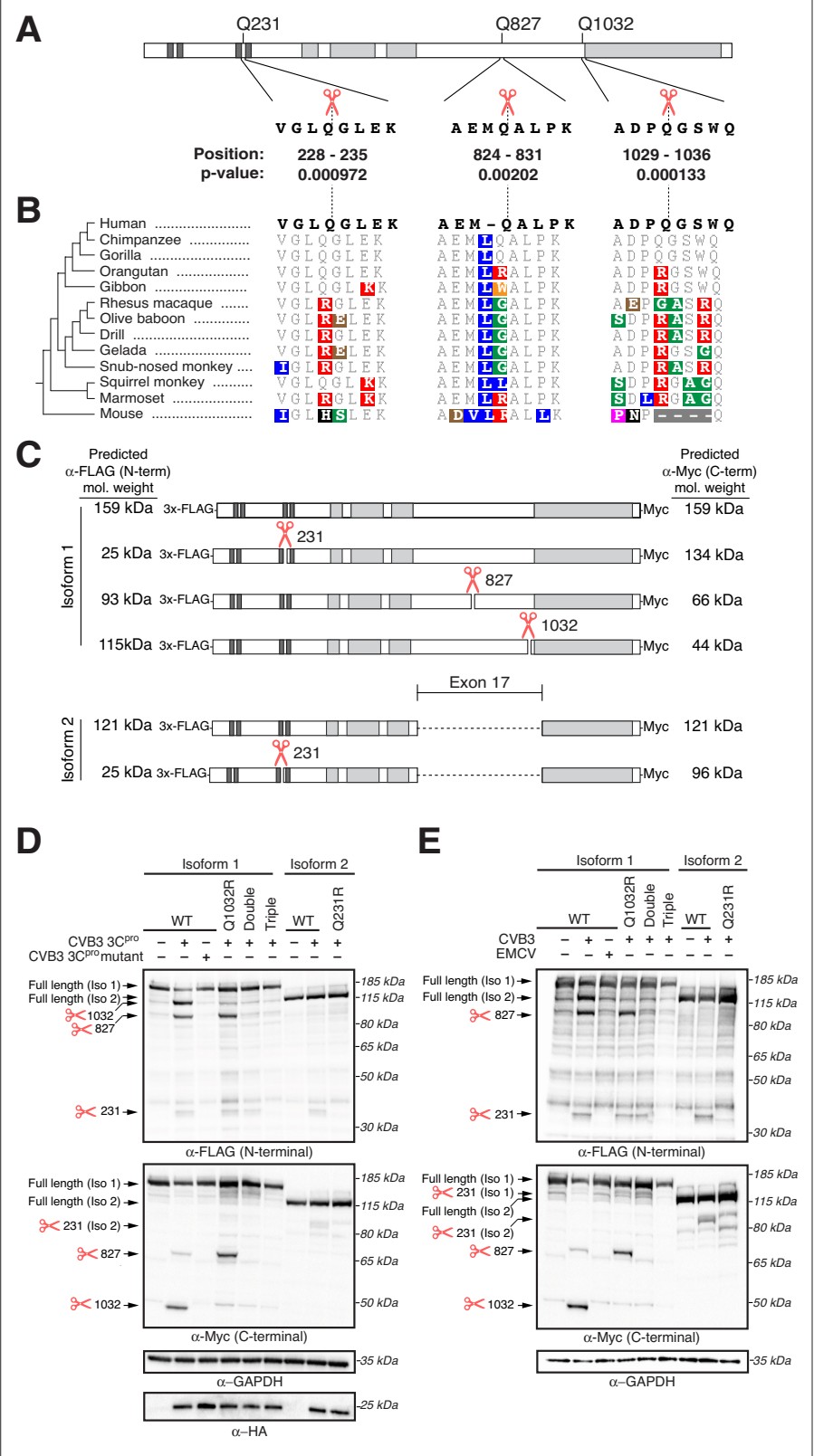

**Figure 5.** Ninein-like (NINL) is cleaved at species-specific sites by virally encoded proteases. (**A**) Schematic of human NINL, with positions of predicted 3C[pro] cleavage sites annotated. Shown are the four amino acids on each side of the predicted cleavage site in human NINL, along with the residue positions and cleavage score predicted using a motif search with the consensus enterovirus cleavage site (see 'Materials and methods').

*Figure 5 continued on next page*

*Figure 5 continued*

(**B**) NINL sequences from 12 primate species and mice for each predicted 3C^pro cleavage site. Amino acid changes relative to human NINL are highlighted in colors to denote differences in polarity and charge. (**C**) Schematic of 3xFLAG-NINL-Myc isoform 1 and isoform 2 constructs, with predicted molecular weights for both amino-terminal (FLAG) and carboxy-terminal (Myc) products upon cleavage by 3C^pro. (**D**) Immunoblots of extracts from HEK293T cells co-transfected with the indicated NINL constructs and either CVB3 3C^pro or the catalytically inactive (C147A) CVB3 3C^pro (mutant). Immunoblots were probed with anti-FLAG (NINL amino-terminus), anti-Myc (NINL carboxy-terminus), anti-HA (3C^pro), and anti-GAPDH (loading control). Arrows to the left of each immunoblot indicate full-length products as well as products corresponding to cleavage at the indicated amino acid residue. Protein molecular weight markers are shown in kilodaltons (kDa) to the right of each immunoblot. Representative images from three biological replicates are shown. (**E**) Immunoblots of extracts from HEK293T cells transfected with the indicated amino-terminal FLAG and carboxyl-terminal Myc tagged NINL constructs and infected with either CVB3 or EMCV (500,000 PFU/mL, MOI ≈ 1.0 for 8 hr). Immunoblots were probed with anti-FLAG (NINL amino-terminus), anti-Myc (NINL carboxy-terminus), and anti-GAPDH (loading control). Arrows to the left of each immunoblot indicate full length products as well as products corresponding to CVB3 3C^pro cleavage at the indicated amino acid residue. Protein molecular weight markers are shown in kilodaltons (kDa) to the right of each immunoblot. Representative images from three biological replicates are shown.

The online version of this article includes the following source data and figure supplement(s) for figure 5:

**Source data 1.** Full raw unedited images for *Figure 5D and E*.

**Figure supplement 1.** 3C and 3CL proteases from diverse viruses cleave ninein-like (NINL) at redundant and unique sites.

**Figure supplement 1—source data 1.** Full raw unedited images for *Figure 5—figure supplement 1A and B*.

---

cleaves NINL at multiple sites, and EMCV, a virus that encodes a 3C^pro that only weakly cleaves NINL at a single site in the N-terminus (*Figure 5—figure supplement 1A*). Consistent with the results we obtained with transfected 3C^pros, we observed cleavage of NINL at species-specific sites 231, 827, and 1032 when we infected with CVB3, and little to no cleavage upon EMCV infection (*Figure 5E*). These data further support that NINL is a target of viral antagonism upon infection in a manner that is both host- and virus-specific.

## Viral proteases disrupt NINL trafficking function

As NINL is a dynein activating adaptor, we next sought to investigate whether proteolytic cleavage of NINL could interfere with cargo trafficking. NINL is well known as a centrosome-associated protein and may also be involved in trafficking endo/lysosomal membranes (*Bachmann-Gagescu et al., 2015*; *Xiao et al., 2021*). In addition, a number of NINL-interacting proteins have been described (*Bachmann-Gagescu et al., 2015*; *Casenghi et al., 2003*; *Dona et al., 2015*; *Kersten et al., 2012*; *Redwine et al., 2017*; *van Wijk et al., 2009*). However, in the context of the interferon response, we have not yet identified a NINL cargo. Thus, we chose to reconstitute NINL's role in dynein-mediated microtubule transport using a heterologous approach (*Kapitein et al., 2010*; *Passmore et al., 2021*). This well-established method uses an inducible heterodimerization system (*Figure 6A*) to induce the movement of normally immotile peroxisomes by recruiting dynein via an activating adaptor to the peroxisome (*Htet et al., 2020*; *Huynh and Vale, 2017*; *Wang et al., 2019*). Briefly, a rapamycin-binding FKBP domain was targeted to peroxisome membranes via the peroxisome targeting sequence (PTS1) of human PEX3. Another rapamycin-binding FRB domain was fused to the NINL and the NINL triple mutant constructs. We truncated the NINL constructs at residue 1062 because some activating adaptors are autoinhibited via interactions between their amino- and carboxy-termini (*Liu et al., 2013*; *Terawaki et al., 2015*; *Urnavicius et al., 2015*). Co-transfection of cells with CVB3 3C^pro, PEX3-mEmerald-FKBP, and WT NINL or the uncleavable NINL triple mutant confirmed that WT NINL is cleaved by CVB3 3C^pro, while the NINL triple mutant is not (*Figure 6B*, *Figure 6—figure supplement 1A*). When we introduced these constructs into human U-2 OS cells, taking advantage of their flat epithelial-like morphology that is ideal for imaging, we found that peroxisomes were distributed throughout the cytoplasm (*Figure 6C*, *Figure 6—figure supplement 1B and C*), but redistributed to the centrosome upon the addition of the rapamycin analog, rapalog (which induces dimerization of FRB and FKBP; *Ho et al., 1996*; *Figure 6C–E*). In contrast, when NINL was co-expressed with CVB3 3C^pro, peroxisomes no longer localized to the centrosome (*Figure 6C–E*). However, the uncleavable

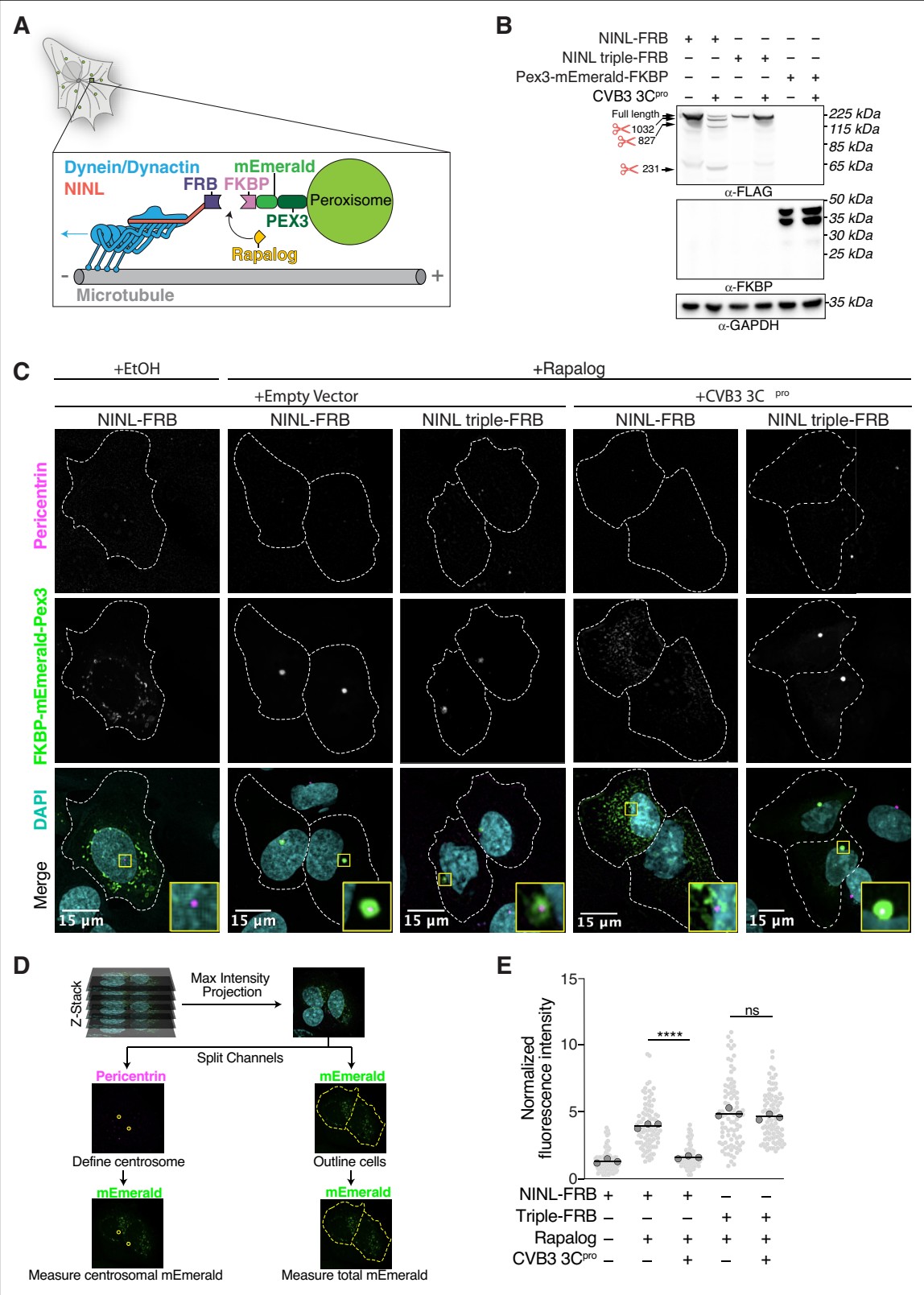

**Figure 6.** CVB3 3C[pro] cleavage of ninein-like (NINL) prevents rapalog-induced dynein-dependent transport of intracellular cargoes. (**A**) Schematic of the peroxisomal trafficking assay. The peroxisomal targeting signal (PTS1) of human PEX3 (amino acids 1–42) was fused to mEmerald and FKBP and a truncated NINL (amino acids 1–1062) was fused to FRB. Dynein-dependent accumulation of peroxisomes at the centrosome, where most minus-ends are located, is initiated by the rapalog-mediated heterodimerization of FKBP and FRB. Blue arrow indicates dynein motility. (**B**) Indicated FRB and

*Figure 6 continued on next page*

*Figure 6 continued*

FKBP constructs transiently expressed with (+) or without (-) the transient co-expression of HA-tagged CVB3 3C^pro in HEK293T cells. Immunoblots were probed with anti-FLAG, anti-FKBP, anti-GAPDH, and anti-HA antibodies. Protein molecular weight markers are shown in kilodaltons (kDa) to the left of each immunoblot. Representative images from three biological replicates are shown. (**C**) Confocal micrographs are displayed as maximum intensity projections of U-2 OS cells, transfected with Pex3-mEmerald-FKBP and the indicated cleavable or uncleavable NINL-FRB fusion constructs with or without the co-expression of CVB3 3C^pro. Where indicated, cells were treated for 1 hr with ethanol (EtOH) as a control or 1 μM rapalog in EtOH prior to fixation. Centrosomes were immunostained with anti-pericentrin and nuclei were visualized with DAPI. 15 μm scale bars indicated in lower-left corner of merged micrographs. Yellow rectangles denote region of cropped inset. Dashed white lines denote cellular boundaries. Representative micrographs from three biological replicates are shown. (**D**) Schematic of the analysis pipeline. (**E**) Quantification of peroxisomal trafficking assays from three biological replicates. The fluorescence intensity of Pex3-mEmerald-FKBP at the centrosome was normalized to the whole-cell fluorescence, and to the areas of the regions of interest used to quantify centrosome versus whole-cell fluorescence. Each datapoint corresponds to an individual cell. The dark gray, large, outlined circles correspond to the mean for each biological replicate. For each condition, n = ~80. The mean across all replicates is denoted by the bold line. Data were analyzed using Kruskal–Wallis with Dunn's post-hoc test for multiple comparisons. ****$p<0.0001$, ns, not significant.

The online version of this article includes the following source data and figure supplement(s) for figure 6:

**Source data 1.** Full raw unedited images for *Figure 6B*.

**Source data 2.** Individual data values for *Figure 6E* and *Figure 6—figure supplement 1C*.

**Figure supplement 1.** Peroxisome distribution remains consistent regardless of presence of CVB3 3C^pro prior to rapalog-induced dynein-dependent transport.

NINL triple mutant was still able to redistribute peroxisomes in the presence of CVB3 3C^pro just as effectively as U-2 OS cells not expressing CVB3 3C^pro (*Figure 6C–E*). Finally, to determine whether viral infection could also disrupt NINL-mediated trafficking, we infected cells with CVB3 following transfection of PEX3-mEmerald-FKBP, and WT NINL or the uncleavable NINL triple mutant. Similar to transfection with viral protease, live CVB3 infection led to a significant reduction in peroxisomes that localized to the centrosome in cells expressing NINL, but not in cells expressing the uncleavable NINL triple mutant (*Figure 7A and B*, *Figure 7—figure supplement 1A and B*). Together, these data demonstrate that site-specific cleavage of NINL by CVB3 3C^pro could disrupt NINL's role in cargo transport.

## Discussion

Pathogenic viruses and their hosts are engaged in genetic conflicts at every step of the viral replication cycle. Each of these points of conflict, which center on direct interactions between viral and host proteins, has the potential to determine the degree to which a virus replicates and causes pathogenesis in a host cell, and the degree to which the immune system can inhibit viral replication. As such, evolutionary adaptation in both host and viral genomes shapes these molecular interactions, leaving behind signatures of rapid evolution that can serve as beacons for points of host–virus interaction (*Daugherty and Malik, 2012*; *Duggal and Emerman, 2012*; *Tenthorey et al., 2022*). Here we use this evolutionary principle to reveal an antiviral role for the dynein activating adaptor NINL. Unique among 36 analyzed dynein, dynactin, and activating adaptor genes, we found that NINL displays a signature of recurrent positive selection in primates. Based on this unusual evolutionary signature in an otherwise highly conserved cellular machine, we hypothesized that NINL may be engaged in an undescribed host–pathogen genetic conflict. Using multiple cell types and knockout clones, we reveal that loss of NINL results in reduced activation of the antiviral innate immune response following IFNα treatment. Consequently, in NINL KO cells several RNA and DNA viruses show significantly increased replication after IFNα pretreatment relative to WT cells. These results indicate that NINL plays an important role in the antiviral immune response.

Further work will be required to determine the mechanistic basis for NINL's antiviral function. The role of activating adaptors in inducing processive dynein motility was only described in 2014 (*McKenney et al., 2014*; *Schlager et al., 2014*). Since that time, the number of established activating adaptors has rapidly expanded, as has our understanding of the molecular interactions between activating adaptors and dynein/dynactin (*Agrawal et al., 2022*; *Cason et al., 2021*; *Chaaban and Carter, 2022*; *Fenton et al., 2021*; *Lau et al., 2021*; *Lee et al., 2020*; *Olenick and Holzbaur, 2019*; *Reck-Peterson et al., 2018*). However, for many activating adaptors, including NINL, much less is known about cargo specificity. Our observation that NINL KO cells have a defect in ISG production following

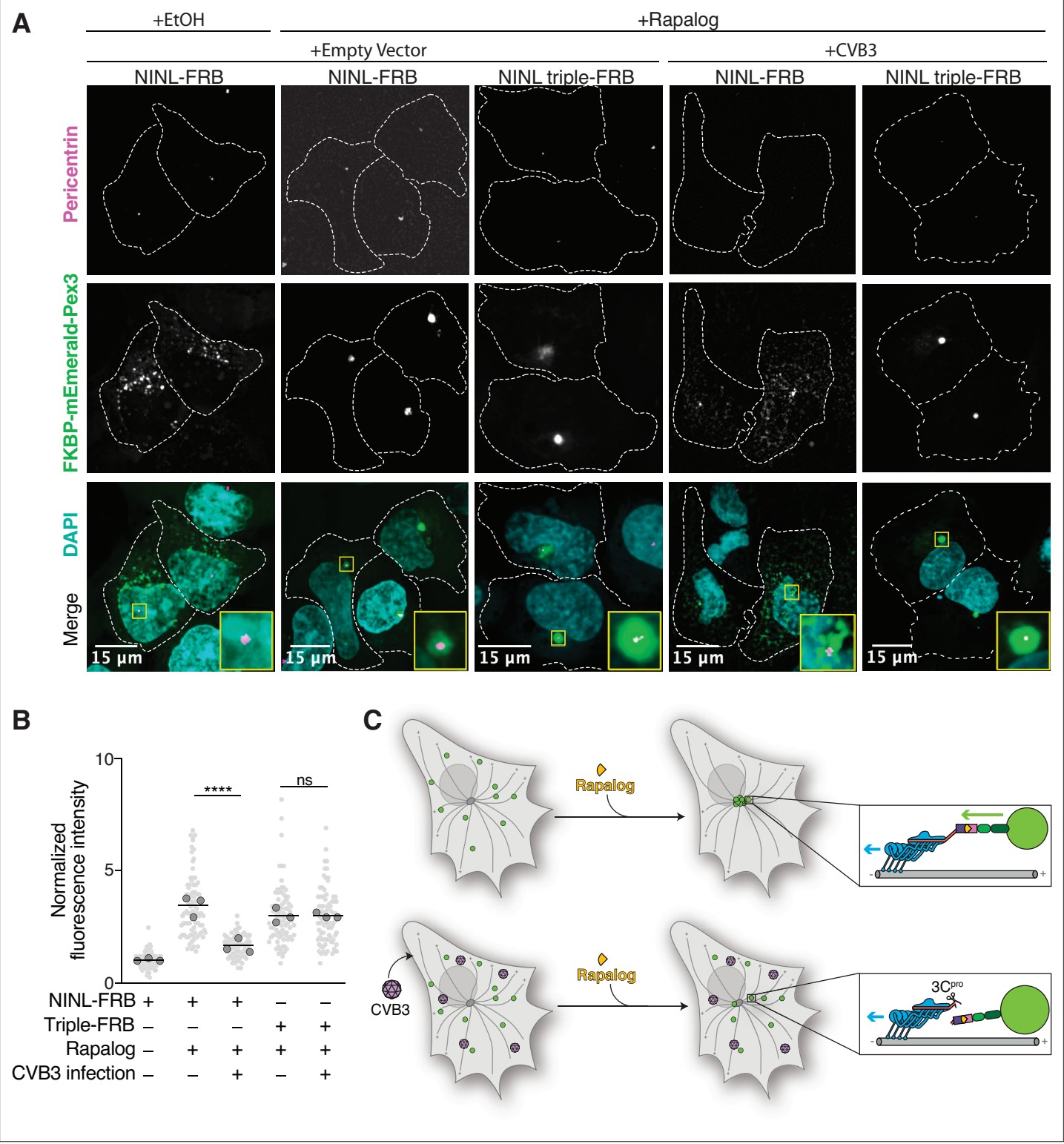

**Figure 7.** Cleavage of ninein-like (NINL) during viral infection prevents dynein-dependent transport of an intracellular cargo. (**A**) Confocal micrographs displayed as maximum intensity projections of uninfected or CVB3 infected U-2 OS cells. Cells were transfected with Pex3-mEmerald-FKBP and the indicated cleavable or uncleavable NINL-FRB fusion constructs, and infected (or mock infected) with CVB3 (500,000 PFU/ml, MOI = ~2) for 5 hr. Cells were then treated for 1 hr with ethanol (EtOH) or 1 µM rapalog prior to fixation. Centrosomes were immunostained with anti-pericentrin and nuclei were visualized with DAPI. 15 µm scale bars are indicated in the lower-left corner of the merged micrographs. Yellow rectangles denote region of cropped inset. Dashed white lines denote cellular boundaries. Representative micrographs from three biological replicates are shown. (**B**) Quantification of

*Figure 7 continued on next page*

*Figure 7 continued*

peroxisomal trafficking assays from three biological replicates. The fluorescence intensity of Pex3-mEmerald-FKBP at the centrosome was normalized to the whole-cell fluorescence, and to the areas of the regions of interest used to quantify centrosome versus whole-cell fluorescence. Each datapoint corresponds to an individual cell and biological replicates can be distinguished by shade. For each condition, n = ~80. The mean across all replicates is denoted by the bold line. Bold circles correspond to the mean for each biological replicate. Data were analyzed using Kruskal–Wallis with Dunn's post-hoc test for multiple comparisons. ****p<0.0001, ns, not significant. (**C**) Schematic of the rapalog-induced pericentrosomal accumulation of peroxisomes and loss of accumulation upon viral infection.

The online version of this article includes the following source data and figure supplement(s) for figure 7:

**Source data 1.** Individual data values for *Figure 7B* and *Figure 7—figure supplement 1B*.

**Figure supplement 1.** Peroxisome distribution remains consistent regardless of CVB3 infection prior to rapalog-induced dynein-mediated transport.

IFNα treatment suggests a role for NINL in the IFN signaling pathway. These results are consistent with recent gene enrichment analyses implicating NINL in several immune pathways, including JAK-STAT signaling (*Chen et al., 2022*). We further showed that despite normal phosphorylation of STAT1 and STAT2 transcription factors following IFNα treatment, pSTAT1 nuclear accumulation is reduced in NINL KO cells, suggesting that NINL is important for the nuclear localization of pSTAT1. However, whether a signaling complex is a direct cargo of NINL, or whether NINL's interaction with dynein and dynactin is required for this function, remain to be determined. Notably, we observe several transcriptional changes in NINL KO cells relative to WT cells, suggesting that NINL plays regulatory roles in the cell beyond our observation of its role in the IFN effect. Like other activating adaptors, understanding the full range of cargos and biological functions of NINL will require additional studies.

Despite the uncertainty of NINL's mechanistic role in the antiviral immune response, we find that several viruses can antagonize NINL function through proteolytic cleavage. Using the model enterovirus, CVB3, we show that the virally encoded 3C protease (3C$^{pro}$) cleaves human NINL at three independent sites, all of which toggle between cleavable and uncleavable even among closely related primates. These changes within the cleavage sites of NINL in primates suggest that virally encoded proteases are one potential evolutionary pressure that is driving the rapid evolution of NINL, similar to other molecular arms races between viral proteases and host proteins (*Tsu et al., 2021b*). Related 3C$^{pros}$ from other picornaviruses, as well as 3CL$^{pros}$ from coronaviruses, also cleave NINL. Intriguingly, even closely related proteases, for instance, 3C$^{pros}$ within the enterovirus clade, have different site preferences within NINL, suggesting that viral protease evolution may be shaping its interactions with NINL. Indeed, among the diversity of picornavirus and coronavirus proteases we tested, we find that NINL cleavage is almost universally maintained despite a wide array of site preferences and cleavage efficiencies. These data indicate that numerous viral proteases convergently cleave NINL, reminiscent of other convergently antagonized targets in the innate antiviral response such as MAVS, TRIF, and NEMO (*Tsu et al., 2021b*). Compellingly, cleavage of NINL by 3C$^{pro}$ during viral infection disrupts the NINL-mediated transport of a heterologous cargo. Along with our data indicating that one function of NINL is to potentiate the innate immune response, these data suggest that cleavage of NINL could be a host-specific mechanism employed by viruses to disrupt the antiviral immune response and promote their own replicative success.

Altogether, our study demonstrates the effectiveness of leveraging genetic signatures of pathogen-driven evolution to identify new components of host innate immunity. Our insights into the species-specific interactions between viruses and NINL provides a glimpse into the impact that viruses may have on the evolution of the intracellular transport machinery and identify a new role for a dynein activating adaptor in the antiviral immune response. These results indicate that components of the otherwise conserved cytoplasmic dynein transport machinery can be engaged in host- and virus-specific interactions and suggest intracellular transport could be an important battleground for host–virus arms races.

# Materials and methods

### Key resources table

| Reagent type (species) or resource | Designation | Source or reference | Identifiers | Additional information |
|---|---|---|---|---|
| Gene (*Homo sapiens*) | NINL | NCBI | NM_025176 | |
| Gene (*Ho. sapiens*) | NIN | NCBI | NM_020921 | |
| Cell line (*H. sapiens*) | HEK293T | ATCC | Cat# CRL-3216; RRID:CVCL_0063 | |
| Cell line (*H. sapiens*) | A549 | ATCC | Cat# CRM-CCL-185; RRID:CVCL_0023 | |
| Cell line (*H. sapiens*) | A549 NINL KO Clone #2 | This paper | | Exon 2 target (TCGGAAACGACCATTTCGCCAGG) |
| Cell line (*H. sapiens*) | A549 NIN KO Clone #3 | This paper | | Exon 5 target (TGGGAAGCGTTACGGACGAAGG) |
| Cell line (*H. sapiens*) | U-2 OS | ATCC | Cat# HTB-96; RRID:CVCL_0042 | |
| Cell line (*H. sapiens*) | U-2 OS NINL KO Clone # 1 | This paper | | Exon 2 target (TCGGAAACGACCATTTCGCCAGG) |
| Cell line (*H. sapiens*) | Flp-In T-REx HCT116 NINL KO clone #13 | This paper | | Exon 6 target (CCACTCGGGTTAAACCGAGCAAG) |
| Cell line (*H. sapiens*) | Flp-In T-REx HCT116 NIN KO clone #2 | This paper | | Exon 3 target (GTTTTGACACGACGGGCACAGGG) |
| Sequence-based reagent | Oligonucleotides | Other | | See *Supplementary file 5* for list of oligonucleotides used in this study |
| Recombinant DNA reagent | pDL2118-mCherry-P2A-3x Flag-NINL-(Q231/827/1032R)-Myc | This paper | | See *Supplementary file 5* for notes on plasmid design |
| Recombinant DNA reagent | pDL2118-mCherry-P2A-3x Flag-NINL(iso1)-Myc | This paper | | See *Supplementary file 5* for notes on plasmid design |
| Recombinant DNA reagent | pDL2118-mCherry-P2A-3x Flag-NINL(iso2 Q231R)-Myc | This paper | | See *Supplementary file 5* for notes on plasmid design |
| Recombinant DNA reagent | pDL2118-mCherry-P2A-3x Flag-NINL(iso2)-Myc | This paper | | See *Supplementary file 5* for notes on plasmid design |
| Recombinant DNA reagent | pDL2118-mCherry-P2A-3x Flag-NINL(Q231R)-Myc | This paper | | See *Supplementary file 5* for notes on plasmid design |
| Recombinant DNA reagent | pDL2118-mCherry-P2A-3x Flag-NINL(Q827R)-Myc | This paper | | See *Supplementary file 5* for notes on plasmid design |
| Recombinant DNA reagent | pDL2118mCherry-P2A-3x Flag-NINL(Q1032R)-Myc | This paper | | See *Supplementary file 5* for notes on plasmid design |
| Recombinant DNA reagent | pQCXIP-HA-CVB3(Nancy Strain)-3Cpro | PMID:33410748 | | |
| Recombinant DNA reagent | pQCXIP-HA-CVB3(Nancy Strain)-3Cpro(C147A) | PMID:33410748 | | |
| Recombinant DNA reagent | pQCXIP-HA-EMCV-3Cpro | PMID:33410748 | | |
| Recombinant DNA reagent | pQCXIP-HA-EV68-3Cpro | PMID:33410748 | | |
| Recombinant DNA reagent | pQCXIP-HA-EV71-3Cpro | PMID:33410748 | | |
| Recombinant DNA reagent | pQCXIP-HA-HepA-3Cpro | PMID:33410748 | | |
| Recombinant DNA reagent | pQCXIP-HA-HRVA-3Cpro | PMID:33410748 | | |
| Recombinant DNA reagent | pQCXIP-HA-NL63-3CLpro | This paper | | See *Supplementary file 5* for notes on plasmid design |
| Recombinant DNA reagent | pQCXIP-HA-Parecho-3Cpro | PMID:33410748 | | |

*Continued on next page*

*Continued*

| Reagent type (species) or resource | Designation | Source or reference | Identifiers | Additional information |
|---|---|---|---|---|
| Recombinant DNA reagent | pQCXIP-HA-PV1-3Cpro | PMID:33410748 | | |
| Recombinant DNA reagent | pQCXIP-HA-Sali-3Cpro | PMID:33410748 | | |
| Recombinant DNA reagent | pQCXIP-HA-SARS1-3CLpro | This paper | | See *Supplementary file 5* for notes on plasmid design |
| Recombinant DNA reagent | pQCXIP-HA-SARS2-3CLpro | This paper | | See *Supplementary file 5* for notes on plasmid design |
| Recombinant DNA reagent | pQCXIP-HA-SARS2-3CLpro(C145A) | This paper | | See *Supplementary file 5* for notes on plasmid design |
| Recombinant DNA reagent | pSPCas9(BB)-2A-Puro-V2.0-NINLexon2-gRNA | This paper | | See *Supplementary file 5* for notes on plasmid design |
| Recombinant DNA reagent | pSPCas9(BB)-2A-Puro-V2.0-NINLexon6-gRNA | This paper | | See *Supplementary file 5* for notes on plasmid design |
| Recombinant DNA reagent | pSPCas9(BB)-2A-Puro-V2.0-NINexon3-gRNA | This paper | | See *Supplementary file 5* for notes on plasmid design |
| Recombinant DNA reagent | pSPCas9(BB)-2A-Puro-V2.0-NINexon5-gRNA | This paper | | See *Supplementary file 5* for notes on plasmid design |
| Recombinant DNA reagent | pSpCas9(BB)-2A-Puro (PX459) V2.0 | Addgene | Cat# 62988; RRID:Addgene_101732 | Gift from Feng Zhang |
| Recombinant DNA reagent | pcDNA3.1(+)-3xFlag-Halo-NINL-Myc-FRB | This paper | | See *Supplementary file 5* for notes on plasmid design |
| Recombinant DNA reagent | pcDNA3.1(+)-3xFlag-Halo_NINL(1-1062)-Myc-FRB | This paper | | See *Supplementary file 5* for notes on plasmid design |
| Recombinant DNA reagent | pcDNA3.1(+)–3xFlag-Halo-NINL(1-1062)(Q231/827/1032R)-Myc-FRB | This paper | | See *Supplementary file 5* for notes on plasmid design |
| Recombinant DNA reagent | pcDNA3.1(+)-Pex3-Emerald-FKBP | This paper | | See *Supplementary file 5* for notes on plasmid design |
| Recombinant DNA reagent | CVB3 (strain Nancy) infectious clone plasmid | PMID:2410905 | | Gift from Dr. Julie Pfeiffer |
| Recombinant DNA reagent | EMCV (strain Mengo) infectious clone plasmid | PMID:2538661 | | Gift from Dr. Julie Pfeiffer |
| Recombinant DNA reagent | T7 plasmid | PMID:31666382 | | Gift from Dr. Julie Pfeiffer |
| Recombinant DNA reagent | SinV (strain Toto1101) infectious clone plasmid | PMID:12388685 | | From Dr. Charles Rice, Rockefeller University |
| Recombinant DNA reagent | VSV-GFP (strain Indiana) | PMID:10400792 | | From Dr. John Rose, Yale University |
| Recombinant DNA reagent | Vaccinia virus (strain Western Reserve) | PMID:12359447 | | From Dr. Richard Condit, University of Florida |
| Recombinant DNA reagent | Vaccinia virus (strain Western Reserve), J3 K175R mutant | PMID:12359447 | | From Dr. Richard Condit, University of Florida |
| Chemical compound, drug | TransIT-X2 | Mirus | MIR 6000 | |
| Chemical compound, drug | Rapalog AP21967 | Takara Bio | 635055 | |
| Peptide, recombinant protein | Recombinant human Interferon alpha 2 | Abcam | ab200262 | |
| Antibody | α-Tubulin antibody (mouse monoclonal) | MilliporeSigma | Cat# T6199; RRID:AB_477583 | IF (1:300) |

*Continued on next page*

*Continued*

| Reagent type (species) or resource | Designation | Source or reference | Identifiers | Additional information |
|---|---|---|---|---|
| Antibody | β-Actin antibody (mouse monoclonal) | Thermo Fisher Scientific | Cat# MA5-15739; RRID:AB_10979409 | WB (1:1000) |
| Antibody | Flag antibody (mouse monoclonal) | Sigma-Aldrich | Cat# A8592; RRID:AB_439702 | WB (1:2000) |
| Antibody | GAPDH antibody (rabbit monoclonal) | Cell Signaling | Cat# 2118; RRID:AB_561053 | WB (1:1000) |
| Antibody | HA antibody (rat monoclonal) | Roche | Cat# 11867423001; RRID:AB_390918 | WB (1:1000) |
| Antibody | IFIT3 antibody (mouse polyclonal) | Abcam | Cat# ab76818; RRID:AB_1566324 | WB (1:1000) |
| Antibody | ISG15 antibody (rabbit polyclonal) | Cell Signaling Technology | Cat# 2743; RRID:AB_2126201 | WB (1:1000) |
| Antibody | Mx1 antibody (rabbit monoclonal) | Cell Signaling Technology | Cat# 37849; RRID:AB_2799122 | WB (1:1000) |
| Antibody | Myc antibody (rabbit monoclonal) | Cell Signaling Technology | Cat# 2278; RRID:AB_490778 | WB (1:1000) |
| Antibody | NIN antibody (rabbit polyclonal) | Lifespan Biosciences | Cat# LS-C668760 | WB (1:1000) |
| Antibody | NINL antibody (rabbit polyclonal) | Thermo Fisher Scientific | Cat# PA5-51438; RRID:AB_2644681 | WB (1:1000) |
| Antibody | Oas1 antibody (rabbit monoclonal) | Cell Signaling Technology | Cat# 14498; RRID:AB_2798498 | WB (1:1000) |
| Antibody | Pericentrin antibody (rabbit polyclonal) | Abcam | Cat# ab4448; RRID:AB_304461 | IF (1:500) |
| Antibody | Phospho Stat1 (Tyr701) antibody (rabbit monoclonal) | Cell Signaling Technology | Cat# 9167; RRID:AB_561284 | WB (1:1000) |
| Antibody | Phospho Stat2 (Tyr690)antibody (rabbit monoclonal) | Cell Signaling Technology | Cat# 88410; RRID:AB_2800123 | WB (1:1000) |
| Antibody | Stat1 antibody (rabbit monoclonal) | Cell Signaling Technology | Cat# 14994; RRID:AB_2737027 | WB (1:1000) |
| Antibody | Stat2 antibody (rabbit monoclonal) | Cell Signaling Technology | Cat# 72604; RRID:AB_2799824 | WB (1:1000) |
| Antibody | Goat anti-mouse IgG (H+L) HRP conjugated (goat polyclonal) | Bio-Rad | Cat# 170-6516; RRID:AB_11125547 | WB (1:10,000) |
| Antibody | Goat anti-mouse IgG (H+L) Alexa Fluor 647 conjugated (goat polyclonal) | Thermo Fisher Scientific | Cat# A21235; RRID:AB_2535804 | IF (1:1000) |
| Antibody | Horse anti-mouse IgG (H+L) HRP conjugated (horse polyclonal) | Cell Signaling Technology | Cat# 7076; RRID:AB_330924 | WB (1:3000) |
| Antibody | Goat anti-rabbit IgG (H+L) Alexa Fluor 568 conjugated (goat polyclonal) | Thermo Fisher Scientific | Cat# A11011; RRID:AB_143157 | IF (1:1000) |
| Antibody | Goat anti-rabbit IgG (H+L) Alexa Fluor 647 conjugated (goat polyclonal) | Thermo Fisher Scientific | Cat# A21247; RRID:AB_141778 | IF (1:1000) |
| Antibody | Goat anti-rabbit IgG (H+L) HRP conjugated (goat polyclonal) | Bio-Rad | Cat# 170-6515; RRID:AB_11125142 | WB (1:10,000) |
| Antibody | Goat anti-rabbit IgG (H+L) HRP conjugated (goat polyclonal) | Cell Signaling Technology | Cat# 7074; RRID:AB_2099233 | WB (1:3000) |

*Continued on next page*

*Continued*

| Reagent type (species) or resource | Designation | Source or reference | Identifiers | Additional information |
|---|---|---|---|---|
| Antibody | Goat anti-rat IgG (H+L) HRP conjugated (goat polyclonal) | Thermo Fisher Scientific | Cat# 31470; RRID:AB_228356 | WB (1:10,000) |
| Commercial assay or kit | RNeasy Plus Mini Kit | QIAGEN | Cat# 74134 | |
| Commercial assay or kit | DNeasy Blood & Tissue Kit | QIAGEN | Cat# 69504 | |
| Software, algorithm | tBlastn | PMID:2231712 | | |
| Software, algorithm | MAFFT | PMID:12136088 | | |
| Software, algorithm | Geneious | Dotmatics | | |
| Software, algorithm | PAML 4 | PMID:17483113 | | |
| Software, algorithm | FEL | PMID:15703242 | | |
| Software, algorithm | MEME | PMID:22807683 | | |
| Software, algorithm | Datamonkey | PMID:29301006 | | |
| Software, algorithm | CHOPCHOP | PMID:27185894 | | |
| Software, algorithm | Salmon | PMID:28263959 | | |
| Software, algorithm | DESeq2 | PMID:25516281 | | |
| Software, algorithm | Scripts for RNAseq analysis | This paper, *Stevens, 2022* | | Available at https://github.com/daugherty-lab/NINL |
| Software, algorithm | Reactome | PMID:31691815 | | |
| Software, algorithm | Scripts for microscopy analysis | This paper, *Stevens, 2022* | | Available at https://github.com/daugherty-lab/NINL |

## Evolutionary analysis

For evolutionary analyses of dynein, dynactin, and activating adaptor genes, UniProt reference protein sequences were used as a search query against NCBI's non-redundant (NR) database using tBLASTn (*Altschul et al., 1990*). For each primate species, the nucleotide sequence with the highest bit score was downloaded and aligned to the human ORF nucleotide sequence using MAFFT (*Katoh et al., 2002*) implemented in Geneious software (Dotmatics; https://www.geneious.com/). Poorly aligning sequences or regions were removed from subsequent analyses. Maximum likelihood (ML) tests were performed with codeml in the PAML software suite (*Yang, 2007*). Aligned sequences were subjected to ML tests using NS sites models disallowing (M7) or allowing (M8) positive selection. The p-value reported is the result of a chi-squared test on twice the difference of the log likelihood (lnL) values between the two models using two degrees of freedom. Analyses were performed using two models of frequency (F61 and F3x4) and both sets of values are reported. For each codon model, we confirmed convergence of lnL values by performing each analysis using two starting omega (dN/dS) values (0.4 and 1.5). For evolutionary analyses of the isolated NINL amino-terminal (dynein/dynactin binding) and carboxy-terminal (cargo binding) regions, the full-length alignment was truncated to only include codons 1–702 or 703–1382, respectively, and PAML analyses were performed as described above.

We used three independent methods to estimate codons within NINL that have been subject to positive selection. PAML was used to identify positively selected codons with a posterior probability greater than 0.90 using a Bayes Empirical Bayes (BEB) analysis and the F61 codon frequency model. The same NINL alignment was also used as input for FEL (*Kosakovsky Pond and Frost, 2005*) and MEME (*Murrell et al., 2012*) using the DataMonkey (*Weaver et al., 2018*) server. In both cases, default parameters were used and codons with a signature of positive selection with a p-value of <0.1 are reported. In all cases, codon numbers correspond to the amino acid position and residue in human NINL (NCBI accession NM_025176.6).

## Molecular cloning

For the plasmid-based CRISPR/Cas9-mediated knockout of NIN and NINL, we designed gRNA target sequences with the web tool CHOPCHOP (*Labun et al., 2016*), available at https://chopchop.cbu.uib.

no/, and synthesized oligonucleotides from Eton Biosciences (San Diego, CA). Each oligonucleotide pair was phosphorylated and annealed using the T4 Polynucleotide Kinase (New England Biolabs, Ipswich, MA). Duplexed oligonucleotides were ligated into BbsI (New England Biolabs) digested pSpCas9(BB)-2A- Puro (pX459) V2.0, a gift from Feng Zhang (Addgene plasmid #62988), using the Quick Ligase kit (New England Biolabs). For cleavage assays, the coding sequence of human NINL isoform 1 (NCBI accession NM_025176.6) was subcloned from the previously described pcDNA5/FRT/TO-BioID-NINL-3xFLAG (*Redwine et al., 2017*) and inserted into pcDNA5/FRT/TO with as part of the following cassette: mCherry-P2A-3xFLAG-NINL-Myc. NINL mutants (Q1032R, Q827/1032R, Q231/827/1032R), human NINL isoform 2 (NCBI accession NM_001318226.2) and the NINL isoform 2 mutant (Q231R) were mutagenized using the Q5 Site-Directed Mutagenesis Kit (New England Biolabs). The plasmids encoding 3C proteases (coxsackievirus B3 [CVB3] 3C$^{pro}$, catalytically inactive [C147A] CVB3 3C$^{pro}$, enterovirus A71 [EV71] 3C$^{pro}$, poliovirus 1 [PV1] 3C$^{pro}$, enterovirus D68 [EV68] 3C$^{pro}$, human rhinovirus A [HRVA] 3C$^{pro}$, encephalomyocarditis virus [EMCV] 3C$^{pro}$, parechovirus A [Parecho] 3C$^{pro}$, hepatitis A virus [HepA] 3C$^{pro}$, and salivirus A [Sali] 3C$^{pro}$) have been described previously (*Tsu et al., 2021a*). To ensure that 3CL$^{pros}$ have precise amino- and carboxy-termini as a result of self-cleavage, sequences for 3CL proteases (SARS2 3CL$^{pro}$, SARS1 3CL$^{pro}$, and NL63 3CL$^{pro}$), including nine residues from the upstream coding region (nsp4) and downstream coding region (nsp6), were ordered as gBlocks (Integrated DNA Technologies, Coralville, IA; *Supplementary file 5*) and cloned into the pQCXIP backbone flanked by an N-terminal eGFP and a C-terminal mCherry-HA sequence. Catalytically inactive (C145A) SARS2 3CL$^{pro}$ was made using overlapping stitch PCR. For the peroxisome trafficking assay, the peroxisomal membrane-targeting sequence (amino acids 1–42) of human PEX3 (NCBI accession NM_003630) with a carboxy-terminal mEmerald fluorescent protein and FKBP was subcloned from the previously described pcDNA5-PEX3-Emerald-FKBP (*Htet et al., 2020*) and into the pcDNA3.1(+) backbone. 3xFLAG-Halo-NINL(1–1062)-Myc-FRB was synthesized as a gBlock (Integrated DNA Technologies) and cloned into the pcDNA3.1(+) backbone. To generate an uncleavable mutant of this construct, we used sequential Q5 mutagenesis to achieve Q231/827/1032R. Following cloning, all plasmids were verified with whole plasmid sequencing. Plasmids and primers used in this study can be found in *Supplementary file 5*. All newly created plasmids will be made available upon request.

## Transfections

All transfections in this study were performed with TransIT-X2 Transfection Reagent (Mirus Bio, Madison, WI) according to the manufacturer's instructions. Briefly, 18–24 hr prior to transfection the desired cells were plated at an appropriate density such that they would be ≥80% confluent at time of transfection. TransIT-X2:DNA complexes were formed following the manufacturer's protocol. The TransIT-X2:DNA complexes were then evenly distributed to cells via drop-wise addition and were incubated in a humidified 5% $CO_2$ atmosphere at 37°C for until they were harvested, assayed, or placed into selection as described below.

## Cell lines

All cell lines used in this study were sourced from the American Type Culture Collection (ATCC; Manassas, VA) unless otherwise indicated and maintained in a humidified 5% $CO_2$ atmosphere at 37°C. All cell lines are routinely tested for mycoplasma by PCR kit (ATCC). HEK293T (human embryonic kidney epithelial cells, ATCC CRL-3216), A549 (human alveolar adenocarcinoma cells, ATCC CCL-185), U-2 OS (human epithelial osteosarcoma cells, ATCC HTB-96), BSC40 (grivet kidney epithelial cells, ATCC CRL-2761), Vero (African green monkey kidney epithelial cells, ATCC CCL-81), and BHK-21 (Syrian golden hamster kidney fibroblast cells, ATCC CCL-10) were maintained in complete growth media, which is composed of Dulbecco's Modified Eagle's Medium with 4.5 g/L glucose, L-glutamine, and sodium pyruvate (DMEM; Corning, Manassas, VA) supplemented with 10% (v/v) fetal bovine serum (FBS; Gibco, Grand Island, NY) and 1% (v/v) Penicillin/Streptomycin (PenStrep; Corning). Flp-In T-REx HCT116 (human colorectal carcinoma cells) were a gift from E. Bennett at the University of California San Diego (La Jolla, CA) but originated in the laboratory of B. Wouters at the University of Toronto (Toronto, ON, Canada) and were maintained in complete growth media supplemented with 100 μg/mL Zeocin. Cells are routinely tested for mycoplasma contamination using mycoplasma by

PCR kit (ATCC, Manassas, VA) and kept at low passage to maintain less than 1 year since acquisition or generation.

## CRISPR/Cas9-mediated gene editing

To generate NIN and NINL knock outs in A549, HCT116, and U-2 OS cell lines, the cells were transfected with 250 ng of the pX459 vector containing the appropriate gRNAs. Transfected cells were enriched 48 hr post-transfection by culturing them with complete growth media supplemented with 1 µg/mL puromycin for 48 hr and then were allowed to recover for 24 hr in complete growth media without puromycin. Following enrichment of transfected cells, monoclonal cell lines were obtained by expanding single-cell clones isolated by limiting dilution. The resulting clones were screened via immunoblotting with gene-specific antibodies anti-NINL rabbit polyclonal antibody (Thermo Fisher Scientific, Waltham, MA) and anti-NIN mouse monoclonal antibody (LSBio, Seattle, WA). Clones determined to be knockouts via immunoblotting were screened further to confirm the presence of CRISPR-induced indels in each allele of the targeted gene. Genomic DNA was isolated using the DNeasy Blood & Tissue Kit (QIAGEN, Hilden, Germany) and the target exons were amplified with EconoTaq polymerase (Lucigen, Middleton, WI). The resulting amplicons were subcloned using the TOPO TA Cloning Kit for Sequencing (Thermo Fisher Scientific) and transformed into DH5α-competent cells. Single colonies were picked, and the plasmids were isolated by miniprep (QIAGEN) and sequenced individually using T3 and T7-Pro primers. All newly created cell lines will be made available upon request.

## Immunoblotting

Harvested cell pellets were washed with 1× PBS, and unless otherwise noted, lysed with RIPA lysis buffer: 50 mM 2-amino-2-(hydroxymethyl)propane-1,3-diol (Tris), pH 7.4; 150 mM sodium chloride (NaCl); 1% (v/v) octylphenyl-polyethylene glycol (IGEPAL CA-630); 0.5% (w/v) sodium deoxycholate (DOC); and 0.1% (w/v) sodium dodecyl sulfate (SDS); 1 mM dithiothreitol (DTT); and cOmplete Protease Inhibitor Cocktail (Roche, Basel, Switzerland) at 4°C for 10 min with end-over-end rotation. Lysates were then centrifuged at maximum speed in a 4°C microcentrifuge for 10 min. The supernatants were transferred to new microcentrifuge tubes and supplemented with NuPage LDS sample buffer (Invitrogen, Carlsbad, CA) and NuPage reducing agent (Invitrogen) prior to a 10 min heat denaturation at 95°C. Lysates were resolved on a 4–12% Bis-Tris SDS-PAGE gel (Life Technologies, San Diego, CA), followed by wet transfer to PVDF membranes (Bio-Rad, Hercules, CA) for 4 hr at 85 V using Towbin buffer: 25 mM Tris base, pH 9.2; 192 mM glycine; 20% (v/v) methanol. Immunoblots were blocked with 5% (w/v) blotting grade nonfat dry milk (Apex Bioresearch Products) in TBS-T: 20 mM Tris pH 7.4; 150 mM NaCl, 0.1% Polysorbate 20 (Tween 20) for 1 hr. Primary antibodies were diluted in TBS-T supplemented with 5% (w/v) BSA and rocked overnight. Primary antibody adsorbed membranes were rinsed three times in TBS-T and subsequently incubated with the appropriate HRP-conjugated secondary antibodies. Membranes were rinsed again three times in TBS-T and developed with SuperSignal West Pico PLUS Chemiluminescent Substrate (Thermo Fisher Scientific) on a ChemiDoc MP Imaging System (Bio-Rad) using Imagelab (Bio-Rad) software. Specifications for antibodies are described in *Supplementary file 6*. The ability of Cas9 Control, NINL KO, and NIN KO to respond to IFNα was assayed by first culturing cells in the presence or absence of 1000U IFNα. Then, 18 hr post-treatment with IFNα, the cells were harvested, lysed, and immunoblotted as described above for STAT1, phospho-STAT1 (Tyr701), STAT2, phospho-STAT2 (Tyr690), MX1, IFIT3, OAS1, and ISG15 (*Supplementary file 6*).

## RNAseq and analysis

All experiments for RNAseq were performed with three biological replicates. Total RNA from mock-treated or IFNα-treated cell lines (1000U, 24 hr treatment) was extracted using an RNeasy Plus Mini Kit (QIAGEN) as indicated in the manufacturer's protocol. The Illumina Stranded mRNA prep kit was used to generate dual-indexed cDNA libraries and the resulting libraries were sequenced on an Illumina NovaSeq 6000 instrument. Total RNA was assessed for quality using an Agilent Tapestation 4200, and samples with an RNA Integrity Number (RIN) greater than 8.0 were used to generate RNA sequencing libraries using the TruSeq Stranded mRNA Sample Prep Kit with TruSeq Unique Dual Indexes (Illumina, San Diego, CA). Samples were processed following manufacturer's instructions, starting with 500 ng

of RNA and modifying RNA shear time to 5 min. Resulting libraries were multiplexed and sequenced with 100 basepair (bp) paired end reads (PE100) to a depth of approximately 25 million reads per sample on an Illumina NovaSeq 6000 instrument. Samples were demuxltiplexed using bcl2fastq v2.20 Conversion Software (Illumina). Sequencing reads were quantified with Salmon (*Patro et al., 2017*) in a quasi-mapping-based mode to the reference genome. Read quantifications were imported and differentially expressed genes across experimental conditions were identified using the R package DESeq2 (*Love et al., 2014*). Reactome pathway analysis was performed by inputting the list of genes with significantly lower expression (adjusted p-value ≤ 0.05, $\log_2$-fold change ≤ –1) in NINL KO cells treated with IFNα relative to WT cells treated with IFNα into the 'Analyze Gene List' tool at https://reactome.org/ (*Jassal et al., 2020*). RNA sequencing data have been deposited in GEO under accession code GSE20678.

## Viral stocks

CVB3 and EMCV viral stocks were generated by co-transfection of CVB3-Nancy or EMCV-Mengo infectious clone plasmids with a plasmid expressing T7 RNA polymerase (generous gifts from Dr. Julie Pfeiffer, UT Southwestern, see *Supplementary file 5*) as previously described (*McCune et al., 2020*). The supernatant was harvested, quantified by plaque assay on Vero cells (CVB3) (see below) or TCID50 on HEK293Tcells (EMCV), and frozen in aliquots at −80°C. Wild-type vaccinia virus Western Reserve strain (NCBI accession NC_006998.1) (VacV WT) and the J3 cap1-methyltransferase K175R vaccinia virus mutant (*Latner et al., 2002*) (VacV J3) were gifts from Dr. Richard Condit (University of Florida). VacV was amplified in BHK cells and quantified by plaque assay as described below. VSV-GFP (Indiana strain, gift from Dr. John Rose, Yale University) was amplified in BSC40s and quantified by plaque assay as described below. Sindbis virus (SinV) was generated by electroporation of in vitro transcribed RNA from plasmid SINV TE/5'2J-GFP (strain Toto1101, from Dr. Charles Rice, Rockefeller University) into BHK cells as previously described (*Bick et al., 2003*) and quantified by plaque assay on BHK cells as described below.

## Viral infection and quantification

For quantification of VSV and SinV, cells (as indicated in each experiment) were seeded in 24-well plates and grown overnight, followed by the addition of 2500 plaque-forming units (PFU)/well of VSV or 250,000 PFU/well SinV. Then, 9 hr after infection for VSV or 24 hr after infection for SinV, viral supernatant was harvested from infected cells. The resulting supernatant was serially tenfold diluted in 24-well plates in DMEM containing 10% FBS and overlaid on BHK cells (ATCC) at 80% confluency for 1 hr. Supernatant was removed from cells 60–120 min post-infection and cells were overlaid with complete DMEM media containing 0.8% carboxymethyl cellulose (MilliporeSigma, Burlington, MA). After 24 hr, the overlay was aspirated and the cells were stained with 0.1% Crystal Violet in 20% ethanol, and then destained with 20% ethanol. Viral concentrations were determined by manually counting plaques.

For quantification of CVB3, cells (as indicated in each experiment) were seeded in 24-well plates and grown overnight, followed by the addition of 25,000 PFU/well virus. Then, 24 hr after infection, viral supernatant was harvested from the infected cells, serially tenfold diluted in 12-well plates in DMEM containing 10% FBS and overlaid on Vero cells (ATCC) at 80% confluency for 1 hr. Supernatant was removed from cells 60–120 min post-infection and cells were overlaid with complete DMEM media containing 1% agarose (Fisher Scientific) and 1 mg/mL neomycin (Research Products International, Mount Prospect, IL) to enhance plaque visualization (*Woods Acevedo et al., 2019*). After 48 hr, agarose plugs were washed out with water and the cells were stained with 0.1% Crystal Violet in 20% ethanol, and then destained with 20% ethanol. Viral concentrations were determined by manually counting plaques.

For quantification of VacV WT and VacV J3, cells (as indicated in each experiment) were seeded in 24-well plates and grown overnight, followed by the addition of 25,000 PFU/well virus. Then, 24 hr after infection, cell-associated virus was harvested by freeze–thaw lysis of the infected cells. Following pelleting of cell debris, virus-containing supernatant was serially tenfold diluted in 24-well plates in DMEM containing 10% FBS and overlaid on BSC40 cells (ATCC) at 80% confluency. After 48 hr, the medium was aspirated, and the cells were stained with 0.1% Crystal Violet in 20% ethanol, and then destained with 20% ethanol. Viral concentrations were determined by manually counting plaques.

## Prediction of NINL cleavage sites by enterovirus 3C$^{pro}$

Putative enterovirus 3C$^{pro}$ cleavage sites within human NINL were predicted using a previously generated polyprotein cleavage motif (*Tsu et al., 2021a*) constructed from >500 non-redundant enterovirus polyprotein sequences. A FIMO motif search against human NINL was conducted using a 0.002 p-value threshold, which we previously determined was sufficient to capture of 95% of enterovirus cleavage sites (*Tsu et al., 2021a*). To enrich for cleavage sites that may be species-specific, sites in which there is variability in the P1 or P1′ sites, which are the primary determinants of cleavage specificity (*Tsu et al., 2021a*), are reported.

## NINL protease cleavage assays

HEK293T cells were co-transfected with 100 ng of epitope-tagged human WT NINL, the NINL double mutant (Q827R, Q1032R), the NINL triple mutant (Q231R, Q827R, Q1032R), NINL isoform 2 or the NINL isoform 2 mutant (Q231R) and with 250 ng of HA-tagged protease-producing constructs for 3C$^{pro}$ assays or 5 ng for 3CL$^{pro}$ assays. Then, 24 hr post-transfection, the cells were harvested, lysed in 1× NuPAGE LDS sample buffer (Invitrogen) containing 5% β-mercaptoethanol (Thermo Fisher Scientific) and immunoblotted as described above.

## NINL virus cleavage assays

HEK293T cells were transfected with 100 ng of epitope tagged human WT NINL, the NINL double mutant (Q827R, Q1032R), the NINL triple mutant (Q231R, Q827R, Q1032R), and NINL isoform 2 or the NINL isoform 2 mutant (Q231R). At 24 hr post-transfection, cells were infected with CVB3 or EMCV at a concentration of 250,000 PFU/well. Then, 9 hr post-infection, the cells were harvested, lysed in 1× NuPAGE LDS sample buffer (Invitrogen) containing 5% β-mercaptoethanol (Thermo Fisher Scientific) and immunoblotted as described above.

## Immunofluorescence of peroxisomes

Cells were grown on fibronectin-coated acid-washed #1 glass coverslips. As applicable, cells underwent the desired treatment prior to a brief permeabilization with 300 μL of 0.5% TritonX-100 (MilliporeSigma) in PHEM buffer: 60 mM piperazine-N,N′-bis(2-ethanesulfonic acid) (PIPES), 25 mM 4-(2-hydroxyethyl)-1-piperazineethanesulfonic acid (HEPES), 10 mM ethylene glycol-bis(2-aminoethylether)-N,N,N′,N′-tetraacetic acid (EGTA), and 4 mM magnesium sulfate heptahydrate (MgSO$_4$·7H$_2$O). After 5 min, 100 μL of a 4% (v/v) formaldehyde (Electron Microscopy Sciences, Hatfield, PA) and 0.5% (v/v) glutaraldehyde (Electron Microscopy Sciences) in PHEM solution was added slowly to the cells and allowed to incubate. After 2 min, all buffer was aspirated from the cells and replaced with the same 4% (v/v) formaldehyde and 0.5% (v/v) glutaraldehyde in PHEM solution and incubated for 20 min at 37°C. After this incubation, the cells were washed three times for 5 min each in PHEM-T (PHEM + 0.1% TritonX-100). The cells were then blocked for 1 hr with a 5% secondary-matched serum solution in PHEM supplemented with 30 mM glycine. The blocking solution was then removed and the desired primary antibodies were added and incubated overnight at 4°C. The following day the cells were washed three times for 5 min in PHEM-T and immunostained with the appropriate secondary antibodies for 1 hr at room temperature. The cells were then washed with PHEM-T and counterstained with 4′,6-diamidino-2-phenylindole (DAPI, Biotium, Fremont, CA). The cells and coverslips were mounted on glass slides with Prolong Glass Antifade Mountant (Thermo Scientific). See *Supplementary file 2* for a list of all antibodies.

## Immunofluorescence of pSTAT1

WT, NINL KO, and NIN KO A549 cells were seeded on to fibronectin-coated coverslips and cultured overnight. Subsequently, cells were treated with 1000U IFNα for 1 hr prior to fixation and immunostaining as described above. Specifically, pSTAT1 was immunostained with anti-phosphorylated STAT1 (Y701) rabbit polyclonal antibodies (Cell Signaling Technology) and goat anti-rabbit IgG (H+L) Alexa Fluor-568 (Thermo Fisher Scientific) and counterstained with DAPI prior to mounting. Z-stacks were acquired using a piezo Z stage. Separate image channels were acquired sequentially using band-pass filters for each channel DAPI: 455/50, pSTAT1: 525/50. The analysis of pSTAT1 localization was performed using ImageJ (U. S. National Institutes of Health, Bethesda, MD) and CellProfiler (Broad Institute of MIT and Harvard, Cambridge, MA). In ImageJ, the original, unmodified z-stack images

were processed by a custom-written batch processing ImageJ script to split and store background corrected DAPI and pSTAT1 channels in TIF format (available at https://github.com/daugherty-lab/NINL copy archived at swh:1:rev:a6bb055917ab1139dd262caa534d0a1860e0ca6c; *Stevens, 2022*). Using CellProfiler, the average 3D fluorescence intensity of total pSTAT1, nuclear pSTAT1, and cytoplasmic pSTAT1 were measured. Briefly, the pSTAT1 channel images were thresholded by the 'Robust Background' method and the DAPI images were thresholded by the 'Minimum cross entropy' method. Using the pSTAT1 threshold, a mask for 'Total pSTAT1' was generated. 'Nuclear pSTAT1' was created by calculating the logical AND of the pSTAT1 threshold image and the DAPI threshold image. 'Cytoplasmic pSTAT1' was generated by calculating the logical AND of the pSTAT1 threshold image and the inverse of the DAPI threshold image. Intensity measurements of pSTAT1 were performed using these three different masks and saved in CSV format (available at https://github.com/daugherty-lab/NINL copy archived at swh:1:rev:a6bb055917ab1139dd262caa534d0a1860e0ca6c, *Stevens, 2022*). The ratio of the mean of pSTAT1 nuclear fluorescence intensity to the mean of pSTAT1 cytoplasmic fluorescence intensity was calculated and plotted using Prism8 (GraphPad, San Diego, CA).

## Confocal microscopy

Cells were imaged using a CSU-W1 spinning disk confocal scanner unit (Yokogawa Electric Corporation, Musashino, Tokyo, Japan) coupled to a six-line (405 nm, 445 nm, 488 nm, 514 nm, 561 nm, and 640 nm) LUN-F-XL laser engine (Nikon Instruments Incorporated, Melville, NY). Emission light from the DAPI, Alexa Fluor 561, and Alexa Fluor 647 was filtered using a quad primary dichroic (405/488/568/647 nm; Semrock, Rochester, NY) and individual bandpass emission filters mounted within the W1 scan head for each channel (450/50, 595/50, and 700/70; Chroma Technology Corporation, Bellows Falls, VT). The W1 was mounted on a Nikon Ti2-E and an Apo TIRF 60× 1.49 NA objective was used to collect images. Image stacks were acquired using a piezo Z-insert (Mad City Labs, Madison, WI). Illumination and image acquisition was controlled by NIS Elements Advanced Research software (Nikon Instruments Incorporated).

## Peroxisome trafficking assay

For imaging of peroxisome accumulation at the centrosome in the presence or absence of 3C[pro] or CVB3 infection, 25,000 U-2 OS cells were plated on fibronectin-coated coverslips and incubated overnight. For 3C[pro] transfected experiments, cells were transfected with the PEX3-Emerald-FKBP construct and either the cleavable NINL-FRB construct or the uncleavable NINL triple mutant construct with or without co-transfection of CVB3 3C[pro]. Then, 18 hr after transfection, the cells were treated with or without 1 µM rapalog (Takara Bio) for 1 hr prior to fixation. For CVB3 infections experiments, the cells were only transfected with the PEX3-Emerald-FKBP construct and either the cleavable NINL-FRB construct or the uncleavable NINL triple mutant construct. Then, 18 hr after transfection, cells were infected with 250,000 PFU (MOI ~2) or mock infected. And 5 hr later, cells were treated with or without 1 µM rapalog for 1 hr (for a total of 6 hr of infection) prior to fixation. Cells from both 3C[pro] experiments and CVB3 infection experiments were fixed and immunostained as described above. Specifically, the centrosome was immunostained with anti-pericentrin rabbit polyclonal antibodies, goat anti-rabbit IgG (H+L) Alexa Fluor-647 (Thermo Fisher Scientific), and counterstained with DAPI prior to mounting. Z-stacks were acquired using a piezo Z stage. Separate image channels were acquired sequentially using bandpass filters for each channel DAPI: 455/50; PEX3-Emerald-FKBP: 525/50; pericentrin: 705/75.

Max intensity projections of Z-stacks were created in Fiji for each separate channel to quantify the peroxisome accumulation at the centrosome. The brightest pericentrin puncta in the 647 channel was identified as the centrosome, and a 60 pixel-wide circle was drawn around it to create a region of interest (ROI). A whole cell ROI was then manually drawn by adjusting the brightness/contrast module's 'Maximum' slider to saturate cellular boundaries. The fluorescence intensity at the centrosome and throughout the cell was then quantified by applying each ROI to the PEX3-Emerald-FKBP/488 channel. The percentage of total fluorescence present at the centrosome was calculated by dividing the intensity of fluorescence at the centrosome by the intensity of fluorescence throughout the cell. The area of the centrosome ROI was then divided by the area of the whole cell ROI to calculate the percentage of the cell's area that the centrosome ROI comprised. The fluorescence intensity

ratio was then divided by the area ratio and plotted using GraphPad Prism. Kruskal–Wallis with Dunn's post-hoc test for multiple comparisons was performed using GraphPad Prism8 (GraphPad).

## Acknowledgements

We thank the members of the Daugherty and Reck-Peterson laboratories, as well as Patrick Mitchell, and Alistair Russell, for helpful suggestions and comments on the manuscript. We also thank the Nikon Imaging Center at UC San Diego and Dr. Eric Griffis for advice on imaging and analysis.

## Additional information

### Competing interests

Samara L Reck-Peterson: Reviewing editor, *eLife*. The other authors declare that no competing interests exist.

### Funding

| Funder | Grant reference number | Author |
|---|---|---|
| National Institute of General Medical Sciences | GM133633 | Matthew D Daugherty |
| National Institute of General Medical Sciences | GM141825 | Samara L Reck-Peterson |
| National Institute of General Medical Sciences | GM007240 | Donté Alexander Stevens |
| Howard Hughes Medical Institute | | Samara L Reck-Peterson |
| National Science Foundation | GRFP | Donté Alexander Stevens |
| Howard Hughes Medical Institute | Gilliam Fellowship | Donté Alexander Stevens |
| Pew Charitable Trusts | Biomedical Scholars Program | Matthew D Daugherty |
| Burroughs Wellcome Fund | Investigators in the Pathogenesis of Infectious Diseases | Matthew D Daugherty |

The funders had no role in study design, data collection and interpretation, or the decision to submit the work for publication.

### Author contributions

Donté Alexander Stevens, Christopher Beierschmitt, Conceptualization, Resources, Formal analysis, Validation, Investigation, Visualization, Writing – original draft, Writing – review and editing; Swetha Mahesula, Resources, Validation, Investigation, Writing – review and editing; Miles R Corley, Formal analysis, Validation, Writing – review and editing; John Salogiannis, Conceptualization, Resources, Writing – review and editing; Brian V Tsu, Conceptualization, Software, Formal analysis, Writing – review and editing; Bryant Cao, Formal analysis, Writing – review and editing; Andrew P Ryan, Resources, Writing – review and editing; Hiroyuki Hakozawki, Software, Validation; Samara L Reck-Peterson, Matthew D Daugherty, Conceptualization, Data curation, Formal analysis, Supervision, Funding acquisition, Validation, Visualization, Writing – original draft, Project administration, Writing – review and editing

### Author ORCIDs

Donté Alexander Stevens ® http://orcid.org/0000-0002-3732-9972
Christopher Beierschmitt ® http://orcid.org/0000-0003-0151-1091
Brian V Tsu ® http://orcid.org/0000-0003-0268-8323
Andrew P Ryan ® http://orcid.org/0000-0002-2630-9837

Samara L Reck-Peterson http://orcid.org/0000-0002-1553-465X
Matthew D Daugherty http://orcid.org/0000-0002-4879-9603

**Decision letter and Author response**
Decision letter https://doi.org/10.7554/eLife.81606.sa1
Author response https://doi.org/10.7554/eLife.81606.sa2

## Additional files

**Supplementary files**

• Supplementary file 1. Evolutionary analyses on dynein, dynactin, and activating adaptor genes. Statistics from PAML analyses on individual dynein, dynactin, and activating adaptor genes.

• Supplementary file 2. Codon positions in ninein-like (NINL) predicted to be evolving under positive selection. NINL codon positions and probability scores from PAML, FEL, and MEME analyses.

• Supplementary file 3. Evolutionary analysis of N-terminal and C-terminal domains of ninein-like (NINL). Statistics from PAML analyses on NINL full length, N-terminal domain only, and C-terminal domain only.

• Supplementary file 4. Differentially regulated transcripts in wild-type (WT) A549, ninein-like (NINL) knockout (KO), and NIN KO cell lines induced and uninduced with interferon (IFN). Results from DESeq comparison from RNAseq of indicated cell lines and under different induction conditions.

• Supplementary file 5. List of primers and gBlocks used. Spreadsheet that details the generation or sourcing of all plasmids used throughout this study.

• Supplementary file 6. List of antibodies used for immunoblots and immunofluorescence. Spreadsheet that details each antibody name, manufacturer, catalog number, and dilutions used.

• MDAR checklist

**Data availability**

All data generated or analyzed during this study are included in the manuscript and supporting files. The NCBI nucleotide database (https://www.ncbi.nlm.nih.gov/nucleotide/) was used to collect sequences for human and non-human primate genes shown in Figure 1. Source data files have been provided for Figures 2–7. RNA sequencing data used in Figure 3 have been deposited in GEO under accession code GSE206784.

The following dataset was generated:

| Author(s) | Year | Dataset title | Dataset URL | Database and Identifier |
|---|---|---|---|---|
| Stevens DA, Beierschmitt C, Mahesula S, Corley MR, Salogiannis J, Tsu BV, Cao B, Ryan AP, Reck-Peterson SL, Daugherty MD | 2022 | Antiviral function and viral antagonism of the rapidly evolving dynein activating adapter NINL | https://www.ncbi.nlm.nih.gov/geo/query/acc.cgi?acc=GSE206784 | NCBI Gene Expression Omnibus, GSE206784 |

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
