## [Editor Report]

The findings of a new player in interferon signaling that is both rapidly evolving and antagonized by a viral protease are exciting.

---

## [Decision Letter]

**Decision letter after peer review:**

Thank you for submitting your article "Antiviral function and viral antagonism of the rapidly evolving dynein activating adapter NINL" for consideration by *eLife*. Your article has been reviewed by 2 peer reviewers, and the evaluation has been overseen by a Reviewing Editor and Carla Rothlin as the Senior Editor. The following individual involved in review of your submission has agreed to reveal their identity: Pei-Yong Shi (Reviewer #2).

This study is very important and well presented. However additional clarifications are required as outlined by the reviewers. In addition to the clarifications two additional experiments should be performed. First, the biochemical assay suggested by Reviewer #2. Second, it would be important to show where in the interferon signaling pathway NINL impacts. Epistasis experiments would clarify where NINL works to better understand the findings in the context of antiviral interferon signaling.

*Reviewer #2 (Recommendations for the authors):*

This paper indicates the role of NINL in the IFN signaling pathway. Data shown in this manuscript support the major claims.

1. The term use of "conflict" is a bit ambiguous. According to the context, this word can mean "Co-evolution" (Line 54), "interaction" (Line 105) or "viral exposure" (Line 94).

2. In Figure 2-supplement 1G, the lines indicating cell types may need to be corrected to match the lanes.

3. Figure 4D: It would be better to enlarge the Myc panel to match the lanes to the other three.

4. Tagging affects protease activities or not.

5. What is the indication of the smaller change of NINL isoform 2 upon cleavage by CVB3 3Cpro?

6. Lines 299 to 301: add a figure illustration of the truncation on NINL and the autoinhibition.

7. For the trafficking assay in Figure 5 and 6, did the transportation of peroxisome affect cell proliferation?

8. For the trafficking assay, did you try another viral 3Cpro and corresponding viral infection? The 3Cpro of CVB3 has the highest cleavage activity among the 3Cpros tested in the previous cleavage assay. Although the result may not be as clear-cut, it would be good to be included to see similar peroxisome re-distribution with another 3Cpro-viral infection example.

---

## [Author Response]

This study is very important and well presented. However additional clarifications are required as outlined by the reviewers. In addition to the clarifications two additional experiments should be performed. First, the biochemical assay suggested by Reviewer #3. Second, it would be important to show where in the interferon signaling pathway NINL impacts. Epistasis experiments would clarify where NINL works to better understand the findings in the context of antiviral interferon signaling.

Thank you for the very positive comments on our work. We have amended our text and figures and performed additional experiments where appropriate at the suggestion of the reviewers to improve our paper.

In response to the biochemical assay suggested by Reviewer #2, please see our detailed explanation below. Briefly, we have encountered substantial solubility issues when attempting to purify full-length NINL from heterologous expression systems. As a result, in our manuscript we relied on in-cell cleavage assays with many specificity controls to establish that viral 3C^pro^ activity is both necessary and sufficient for the site-specific cleavage of NINL. Trouble-shooting methods to obtain soluble NINL, which may not even be possible, would be well beyond the revision period that *eLife* aims for.

In response to the suggestion to determine where in the interferon signaling pathway NINL impacts, we have now included new figures (Figure 4 and Figure 4—figure supplement 1), which demonstrate that NINL is required for the robust nuclear localization of pSTAT1. By finding that pSTAT1 nuclear localization is disrupted in NINL KO cells, these data provide a plausible mechanism by which NINL KO negatively impacts ISG production. These data also suggest that NINL facilitates nuclear localization of pSTAT1, which is consistent with its role as a cellular trafficking protein. Future experiments beyond the scope of this paper are required to understand the exact molecular interactions and molecular mechanism of this function of NINL. However, we believe these new data reveal that NINL impacts the interferon signaling pathway at the point of nuclear translocation of pSTAT1, following STAT phosphorylation but prior to ISG transcription. We have illustrated this new conclusion in Figure 4C.

Inclusion of these data prompted us to add Dr. Hiroyuki Hakozawki as an additional author on the manuscript. Dr. Hakozawki helped us with the analysis of the microscopy data shown in Figure 4. All authors approve of the addition and place in the author list.

Reviewer #2 (Recommendations for the authors):This paper indicates the role of NINL in the IFN signaling pathway. Data shown in this manuscript support the major claims.1. The term use of "conflict" is a bit ambiguous. According to the context, this word can mean "Co-evolution" (Line 54), "interaction" (Line 105) or "viral exposure" (Line 94).

We thank the reviewer for pointing this out. We have modified the manuscript throughout to clarify when we are referring genetic conflicts (i.e. two genomes competing against each other) versus specific host-virus molecular interactions.

2. In Figure 2-supplement 1G, the lines indicating cell types may need to be corrected to match the lanes.

We believe the reviewer is referring to Figure 2—figure supplement 1D. We have adjusted these lines and thank the reviewer for noticing this.

3. Figure 4D: It would be better to enlarge the Myc panel to match the lanes to the other three.

We thank the reviewer for this comment. This change is now reflected in the new Figure 5D.

4. Tagging affects protease activities or not.

In Figure 5 (previously Figure 4), we test the cleavage of NINL by tagged and overexpressed CVB3 3C^pro^ (panel D) compared to cleavage of NINL during CVB3 infection (panel E). In this case, the protease expressed during viral infection (without any epitope tags) cleaves with similar specificity and intensity as tagged overexpressed protease. These results indicate that a tagged 3C^pro^ has similar protease activity as the protease expressed during viral infection. We have not tested the effect of tagging on other 3C^pros^, although we believe this result with CVB3 suggests tagging is unlikely to impact 3C^pro^ activity. Our 3CL^pros^ are expressed as fusion proteins flanked by their natural polyprotein cleavage sites. As a result, after the protease cleaves itself out of the fusion construct, it is untagged since the HA tag remains on the liberated C-terminal mCherry domain.

5. What is the indication of the smaller change of NINL isoform 2 upon cleavage by CVB3 3Cpro?

We are somewhat unclear on what the reviewer is asking for. One possibility is the reviewer is asking why there is little to no decrease in the band intensity of full length NINL isoform 2 after cleavage by CVB3 3C^pro^ relative to the obvious decrease in band intensity of full length NINL isoform 1 after cleavage. If that is the question, then we would point out that isoform 2 lacks the two strongest cleavage sites for CVB3 3C^pro^ (Q827 and Q1032) since the exon (exon 17) containing those sites is spliced out of isoform 2. As a result, the only site remaining in isoform 2 for cleavage by CVB3 3C^pro^ is Q231, which we find to be only weakly cleaved relative to Q827 and Q1032 as observed in the Figure 5D and 5E (previously Figure 4D and 4E). A schematic of this, including positions of cleavage, position of the spliced out exon, and the expected molecular weight of full length and cleaved isoform 1 and isoform 2 are shown in Figure 5C (previously 4C). If the reviewer is instead asking about the size differences in the cleavage products observed in isoform 2 after cleavage, the observed sizes of the cleavage products match those predicted and illustrated in Figure 5C.

6. Lines 299 to 301: add a figure illustration of the truncation on NINL and the autoinhibition.

We have added a schematic in Figure 6 —figure supplement 1 (previously Figure 5 —figure supplement 1) to illustrate the comparison of full length NINL to the construct we used in the peroxisome assay as the reviewer suggested. We did not include information about the autoinhibition because it has not been formally shown that the carboxy-terminal region of NINL is autoinhibitory. The decision to truncate NINL at amino acid 1062 was made based on the fact that some dynein activating adaptors are regulated by autoinhibition by carboxy-terminal coiled coils. For example, the BICD family of dynein activating adaptors have a carboxy-terminal region that folds back on itself and interacts with the amino-terminal region, occluding the dynein/dynactin binding interface. While we have preliminary data that NINL could exist in a similar autoinhibited state, further studies outside of the scope of this paper are required to prove that autoinhibition is occurring with NINL.

7. For the trafficking assay in Figure 5 and 6, did the transportation of peroxisome affect cell proliferation?

The induced peroxisome motility assays are done in a time frame of one hour prior to fixation. This timeframe is too short for us to determine the effect of peroxisome motility on cell proliferation, which would require days. Based on this timing, we also believe that any effects we see in the peroxisome assay are not confounded by any potential cell proliferation effects.

8. For the trafficking assay, did you try another viral 3Cpro and corresponding viral infection? The 3Cpro of CVB3 has the highest cleavage activity among the 3Cpros tested in the previous cleavage assay. Although the result may not be as clear-cut, it would be good to be included to see similar peroxisome re-distribution with another 3Cpro-viral infection example.

Our intention with the inducible peroxisome trafficking assay was to establish that site-specific cleavage of NINL by a viral protease could disrupt NINL’s role in cargo transport. As the reviewer mentions, CVB3 3C^pro^ was ideal for these experiments as we show that it has some of the highest cleavage activity of the proteases tested. Additionally, we have extensively mapped and verified the site specificity of CVB3 3C^pro^ over the NINL amino acid sequence, allowing us to generate uncleavable mutants that serve as essential controls. As noted in Figure 5—figure supplement 1A (previously Figure 4—figure supplement 1A), the cleavage position was variable amongst the picornaviral 3C^pros^ tested. Thus, to perform the suggested experiment, we would need to iteratively map, clone, and verify NINL mutants resistant to cleavage by specific viral 3C^pros^, something that we feel is beyond the scope of *eLife’s* suggested revision time.